# Loop-extruders alter bacterial chromosome topology to direct entropic forces for segregation

Janni Harju [1], Muriel C. F. van Teeseling[2] & Chase P. Broedersz [1,3] ✉

Entropic forces have been argued to drive bacterial chromosome segregation during replication. In many bacterial species, however, specifically evolved mechanisms, such as loop-extruding SMC complexes and the ParABS origin segregation system, contribute to or are even required for chromosome segregation, suggesting that entropic forces alone may be insufficient. The interplay between and the relative contributions of these segregation mechanisms remain unclear. Here, we develop a biophysical model showing that purely entropic forces actually inhibit bacterial chromosome segregation until late replication stages. By contrast, our model reveals that loop-extruders loaded at the origins of replication, as observed in many bacterial species, alter the effective topology of the chromosome, thereby redirecting and enhancing entropic forces to enable accurate chromosome segregation during replication. We confirm our model predictions with polymer simulations: purely entropic forces do not allow for concurrent replication and segregation, whereas entropic forces steered by specifically loaded loop-extruders lead to robust, global chromosome segregation during replication. Finally, we show how loop-extruders can complement locally acting origin separation mechanisms, such as the ParABS system. Together, our results illustrate how changes in the geometry and topology of the polymer, induced by DNA-replication and loop-extrusion, impact the organization and segregation of bacterial chromosomes.

Many bacteria contain a single circular chromosome that is simultaneously replicated and segregated during the cell cycle[1–4]. The physical segregation of these large DNA polymers (~1 mm) must be achieved rapidly, accurately, robustly, and within the tight confinement of the cell (~1 μm), to ensure the viability of daughter cells. Various mechanisms, including entropic forces[5], loop-extruding Structural Maintenance of Chromosomes (SMC) complexes[6–9], and the origin segregating ParABS system[10–12], are implicated in the bacterial chromosome segregation process. However, the interplay between and the relative importance of these mechanisms are not fully understood. In

this study, we investigate when in the replication cycle, where along the chromosome, and how these different components can individually and collectively contribute to bacterial chromosome segregation.

The concept of entropic segregation forces arises from the field of polymer physics. Put simply, two spatially confined polymers with excluded volume interactions can segregate because there are more configurations that they can adopt when they do not co-localize[5,13,14]; segregated states are more numerous and therefore have higher entropy. Entropic forces can also cause chromosomal loops to segregate, which has been proposed to affect the organization and

[1]Department of Physics and Astronomy, Vrije Universiteit Amsterdam, Amsterdam, The Netherlands. [2]Junior research group Prokaryotic Cell Biology, Department for Microbial Interactions, Institute of Microbiology, Friedrich-Schiller-Universität, Jena, Germany. [3]Arnold Sommerfeld Center for Theoretical Physics and Center for NanoScience, Department of Physics, Ludwig-Maximilian-University Munich, Munich, Germany. ✉e-mail: c.p.broedersz@vu.nl

segregation of eukaryotic[15] and prokaryotic[16] chromosomes. Previous work has suggested that entropic segregation forces could largely explain bacterial chromosome segregation[5,17]. These works theoretically focused on the entropic segregation of two fully replicated chromosomes, whereas in vivo, segregation is concurrent with replication[18,19]. In simulations, concurrent chromosome replication and segregation could only be achieved if a "concentric-shell" model was used, where one replicated chromosomal strand was given a larger accessible volume than the other[17]. A later simulation study also found that purely entropic forces are insufficient for segregation, but that segregation can be achieved if chromosomal regions corresponding to *E. coli* Macrodomains are geometrically constrained[20]. Hence, while the theoretical concept of entropic segregation has gained traction, the role and importance of entropic segregation forces are still subjects of debate. When in the replication cycle are entropic segregation forces effective? And how do they act in conjunction with dedicated chromosome segregation mechanisms?

Three widely spread biological mechanisms are known to contribute to chromosome segregation in prokaryotes: loop-extrusion by SMC complexes[9,21,22], terminus segregation by the translocase FtsK[23–26], and origin segregation by the ParABS system[27]. SMC complexes are motor proteins that attach onto the chromosome, and then progressively reel in a chromosomal loop in an ATP-dependent process. However, it remains unclear how loop-extrusion biophysically promotes bacterial chromosome segregation[3]. FtsK "pumps" replicated terminal regions in opposite directions right before cell division[28,29]. Finally, at least in the model organism *Caulobacter crescentus*, the ParABS system can be modeled as a locally acting force that tethers one origin of replication to a pole, and pulls the other origin to the opposite pole[30–34]. In *C. crescentus*, the ParABS system and FtsK are essential for chromosome segregation[12,26] whereas the SMC condensin does not appear to be required[35–37]. By contrast, *Escherichia coli* has no ParABS system, and instead relies on loop-extrusion and FtsK for faithful chromosome segregation[6,23,38]. Finally, in *Bacillus subtilis*, the ParABS system and FtsK only become critical for chromosome segregation in the absence of condensin[39–41]. These observations suggest that biological mechanisms can play important but sometimes overlapping roles during bacterial chromosome segregation. The relative contributions of these mechanisms to chromosome segregation in various species, as well as their interplay with entropic forces, remain unclear.

Here, we start by revisiting the theoretical question of whether entropic forces alone can segregate bacterial chromosomes. We develop a biophysical model and simulations for entropic segregation that take the geometry and topology of a replicating chromosome into account. We show that, until late replication stages, purely entropic forces cause the alignment of replicated chromosome strands, and therefore actually inhibit chromosome segregation. Remarkably however, our results also demonstrate that origin-proximally loaded loop-extruders can transform these segregation-inhibiting entropic forces into "topo-entropic" forces that efficiently drive global chromosome segregation. By contrast, we find that locally acting ParABS-like separating forces alone can be insufficient to segregate terminal regions. This suggests that faithful ParABS-based chromosome segregation requires either loop-extrusion or an additional dedicated terminus-segregation mechanism. Our work explains results from existing knock-out experiments, makes novel, testable predictions, and provides a conceptual framework for understanding how different bacterial chromosome segregation mechanisms operate in unison.

## Results

### Topo-entropic segregation model
We develop a biophysical model to gain physical intuition for how entropic forces act on the global organization of a replicating bacterial chromosome. In contrast to previous theoretical work[5], we consider how the extent of replication affects the geometry and topology of the chromosome and thereby the direction of entropic forces. For simplicity, we do not include structural features such as Chromosome Interaction Domains (CIDs)[36] or E. coli Macrodomains[20], nor do we assume constraints on the replication fork positions[5,42]. Using this topo-entropic segregation model, we investigate how entropic forces are affected by loop-extruders that are loaded at the origins of replication (*ori*) and traverse along the chromosomal arms towards the terminus (*ter*), as observed in multiple bacterial species.

### Entropy does not segregate partially replicated circular chromosomes
To describe the global organization of a replicating chromosome, we can employ a coarse-grained polymer model. In this model, the chromosome is represented as a circular polymer with $N$ coarse-grained monomers of length $b$—set to be larger than the persistence length of DNA—at a time point where $R$ monomers have been replicated. This partially replicated chromosome exhibits excluded volume interactions and is confined to a cylinder of diameter $d$ and length $L$, representing the nucleoid. Chromosomes are highly compressed ($Nb^{1/v} \gg L^{1/v}$, where $v$ is the Flory exponent): without confinement, they expand drastically[43–45]. In a cellular confinement, we can describe chromosomal strands as compressed entropic springs (Fig. 1A, B), which tend to extend along the long axis of the cell if the cell is sufficiently long[5,17,46].

To find the entropically preferred chromosome configuration at a given replication stage, we first decompose the chromosome into the largest possible ring and a remaining linear segment connecting the replication forks (Fig. 1C). In the first half of the replication cycle ($R < N/2$), the largest ring is given by a union of one replicated strand and the unreplicated strand of the chromosome, together of length $N$. Once more than half of the chromosome has been replicated ($R > N/2$), the largest ring is given by the union of the two replicated strands, together of length $2R$. We seek long-axis configurations that maximize the entropies of the ring and the linear segment, under the constraint that they are conjoint at the replication forks. To simplify our reasoning, we neglect excluded volume interactions between the ring and the linear segment, as we can use a blob scaling analysis to show that including these interactions does not change the entropically favored configuration of the chromosome (Supplementary Note 1, Supplementary Fig. 1).

The configurational entropy of the largest ring is optimized when it maximally extends across the length $L$ of the nucleoid. But, the ring by itself has no preferred orientation; it can freely rotate. We note that including ParABS-like forces, polymer structures such as Macrodomains[20] or fixed loops[16] could break this rotational symmetry. However, even in the absence of such effects, the remaining linear segment of the partially replicated chromosome breaks this symmetry and determines the entropically preferred orientation of the chromosome.

In the first half of the replication cycle, once the newly replicated linear segment is long enough to be compressed ($R \gg (d/b)^{1/v}$), the linear segment entropically prefers to orient and extend along the long-axis of the cell. The replication forks are hence pushed apart, resulting in a "fork-segregated", left-*ori*-right configuration, where the newly replicated chromosome segments lie parallel to each other (Fig. 1D, first cell). For the first half of the cell cycle, purely entropic forces thus inhibit chromosome segregation.

In the second half of the replication cycle, as long as the unreplicated linear segment is long enough to be compressed ($N - R \gg (d/b)^{1/v}$), the linear segment's entropy is again larger when the replication forks are pushed apart along the long-axis of the cell. The chromosome hence still favors a fork-segregated orientation (Fig. 1D, second cell).

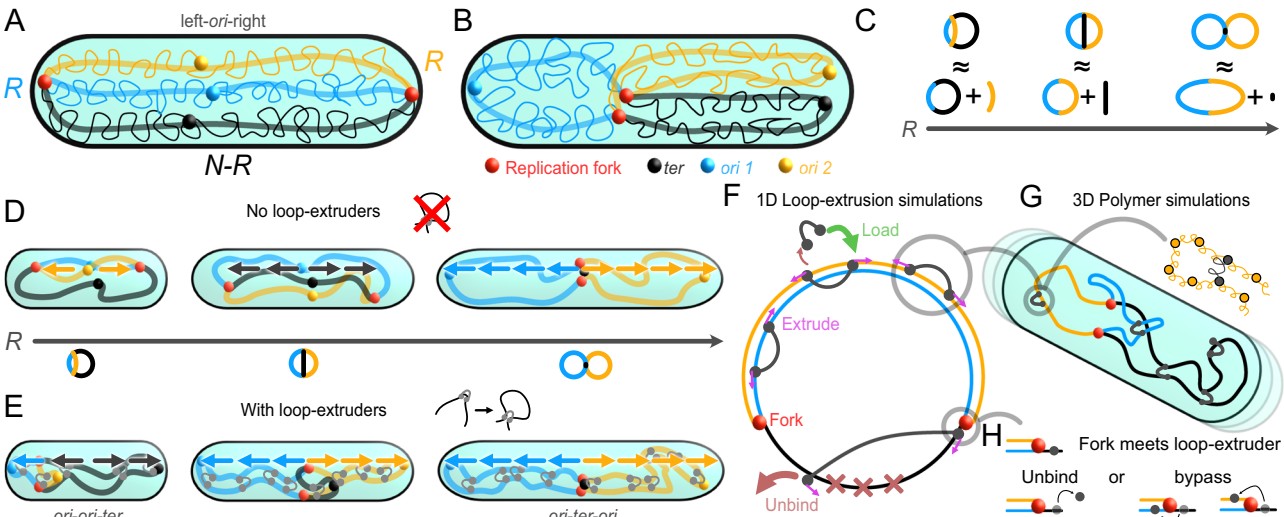

**Fig. 1 | Schematics for topo-entropic segregation model and simulations. A** A highly compressed circular chromosome of $N$ monomers, with $R$ monomers replicated, confined to a cylinder. Orange and blue lines depict newly replicated chromosomal strands, black line the unreplicated strand. In schematics, we depict only the orientation of each strand (thick lines), rather than the microscopic configuration of the highly compressed polymers (thin wiggly lines). Filled circles mark *ori*s (blue and orange), replication forks (red) and the *ter* (black). **B** Similar to A, but showing a sister strand-segregated configuration. **C** At each replication stage, we decompose the partially replicated chromosome into the largest possible ring and a linear segment. **D** The entropically favored global orientation for three distinct replication stages. Colored arrows indicate the direction of purely entropic forces. Until late replication stages, we expect fork-segregated left-*ori*-right configurations, and at late replication stages, a segregated *ori*-*ter*-*ori* configuration. **E** When loop-extruders are loaded at *ori*s, the opposite arms of the chromosome are effectively tied together. The entropically favored global configurations under these constraints are sketched at three replication stages. Overlaid arrows indicate the direction of topo-entropic forces. For $R < N/2$, we expect an *ori*-*ori*-*ter* configuration, and for $R \geq N/2$, a segregated *ori*-*ter*-*ori* configuration. **F** Bidirectional loop-extrusion is simulated in 1D[50,51] on a replicating circular chromosome. The replication forks proceed independently, creating new sites where loop-extruders can move. Loop-extruders are preferably loaded at the *ori*s. The terminal region has an enhanced offloading rate. **G** Loop-extruder and replication fork positions from the 1D simulations are used to constrain 3D bead-spring polymer simulations, where loop-extruders act as springs between monomers. The replicating polymer is confined to an exponentially growing cylinder. **H** When a replication fork encounters a loop-extruder leg, the loop-extruder either unbinds or steps onto either strand on the other side of the fork.

Once replication is nearly complete ($N - R \lesssim (d/b)^{1/\nu}$), the unreplicated linear segment becomes so short that it no longer gains entropy by aligning with the long axis, and thus no longer pushes the replication forks apart. This means that the limit of two conjoined ring polymers is reached, and chromosome segregation finally becomes entropically preferable, as previously argued[5] (Fig 1D, third cell).

In summary, our biophysical model shows that entropic forces acting on a partially replicated bacterial chromosome push the replication forks towards opposite sides of the cell. The resulting fork-segregated states allow all three chromosomal strands attached at the replication forks to extend. Our theory hence suggests that entropic forces alone cannot drive bacterial chromosome segregation until late replication stages, and that bacteria must use other mechanisms to achieve concurrent chromosome replication and segregation.

### Loop-extrusion redirects entropic forces enabling segregation during replication

We next consider the effect of bidirectional loop-extruders primarily loaded at the origins of replication, such as condensin in *B. subtilis*[9,47], *Streptococcus pneumoniae*[48], or *C. crescentus*[49]. Importantly, loop-extruders loaded at the same point and moving in the same direction will collectively "zip up" or tie the opposite arms of the chromosome together. This implies that, in the presence of enough loop-extruders, the largest ring at a given replication stage is (at least partially) linearized, with loop-extruder loading sites acting as "chain ends". This linearized largest ring still entropically prefers to extend across the long-axis of the cell, but now in a specific orientation, so that the loop-extruder loading site is near a cell pole. For the first half of the replication cycle, the preferred chromosome configuration is then *ori*-*ori*-*ter* (Fig. 1E, first cell), and for the second half, *ori*-*ter*-*ori* (Fig. 1E, second and third cell). A blob scaling argument shows that loop-extruders loaded

at the origins also shift the free energy due to chain overlap in favor of the global orientations sketched in Fig. 1E (Supplementary Note 2, Supplementary Fig. 1). Conceptually, the entropic forces are modified by the change in the chromosome's effective topology into "topo-entropic" forces. In contrast to purely entropic forces, topo-entropic forces drive chromosome segregation concurrent with replication.

In summary, by considering how replication and constraints imposed by loop-extruders affect the large-scale geometry and topology of a replicating bacterial chromosome, we have developed a topo-entropic segregation model for bacterial chromosome segregation. Unlike previously proposed[5], our model predicts that purely entropic forces should inhibit rather than promote bacterial chromosome segregation for most of the replication cycle. Importantly, we also find that concurrent replication and segregation can be recovered if loop-extruders redirect entropic forces by tying the arms of the circular chromosome together, thereby changing the chromosome's effective topology.

### Simulations of a replicating chromosome

To test our predictions, we construct a computational model for loop-extrusion on a replicating bacterial chromosome, by adapting a previously published algorithm for a single bacterial chromosome[50,51], with system parameters based on experimental data[50–54] (Supplementary Notes 3-4, Supplementary Table 1). For simplicity, we do not consider possible effects of transcription on loop-extrusion[55,56], since previous simulation work suggests that although collisions with RNA polymerases can slow down bacterial condensins, this does not prevent loop-extruders from zipping up the chromosomal arms[50]. Briefly, we first simulate the movements of a set of loop-extruders and two independently moving replication forks on a 1D lattice (Fig. 1F). In eukaryotes, replication machinery have been proposed to act as

(permeable) road-blocks to loop-extruding cohesins[55,57], but less is known about how bacterial SMCs interact with replication forks. We model loop-extruder fork interactions by assuming that when a loop-extruder leg encounters a replication fork, it either unbinds with some probability $P_U$, or steps to either DNA strand on the other side of the fork with equal probabilities (Fig. 1H, Supplementary Fig. 2). We find that increasing $P_U$, which effectively stops loop-extrusion at the replication forks, slightly improves segregation, but does not qualitatively impact our results (Supplementary Fig. 3). We hence use a value of $P_U = 0$ unless otherwise stated. The loop-extruder and fork positions from the 1D simulations are then imposed as moving constraints in molecular dynamics simulations of a 3D polymer (Methods, Fig. 1G). These coarse-grained simulations have sufficiently strong excluded volume effects that the polymer exhibits self-avoiding behavior (Supplementary Note 5, Supplementary Fig. 4), but infrequent strand-passing is allowed. Strand passing can resolve topological entanglements, mimicking the effect of topoisomerases. Finally, the replicating chromosome is confined to an exponentially growing cylinder to model cell growth, and we can add origin-pulling forces to qualitatively model the ParABS system (Supplementary Note 6). This computational model allows us to test how different mechanisms (individually and together) affect the spatio-temporal organization of replicating chromosomes.

## Loop-extruders prevent entropically favored fork segregation

The efficiency of chromosome segregation depends on two factors: the directions of the relevant forces, which determine the steady-state configuration a partially replicated chromosome would relax to given enough time, and the speed at which the system relaxes. To test our predictions for the directions of entropic forces and the resulting global organization with and without loop-extrusion, we start by investigating steady-state simulations with a given, fixed replicated length $R$. We initialize simulations with unsegregated, overlapping *ori-ter* configurations. Simulations are run long enough that the mean long-axis positions of various loci converge to steady-state values (Supplementary Fig. 5). Since the two replicated strands are identical, averaging over sampled configurations yields the same statistics for both replicated strands, as well as for both replication forks. We hence orient all chromosome configurations to break these symmetries (Methods). This orientation allows us to look for indicators of chromosome and fork segregation in the data.

Mean long-axis positions of monomers show that, without loop-extruders, for $N/4 \leq R < N$, the replication forks are on average in different cell halves (Fig. 2A). The replication forks are hence clearly separated along the long-axis of the cell, and the replicated chromosomal strands are mostly aligned in slightly shifted left-*ori*-right configurations. By contrast, with loop-extruders the chromosomes are in

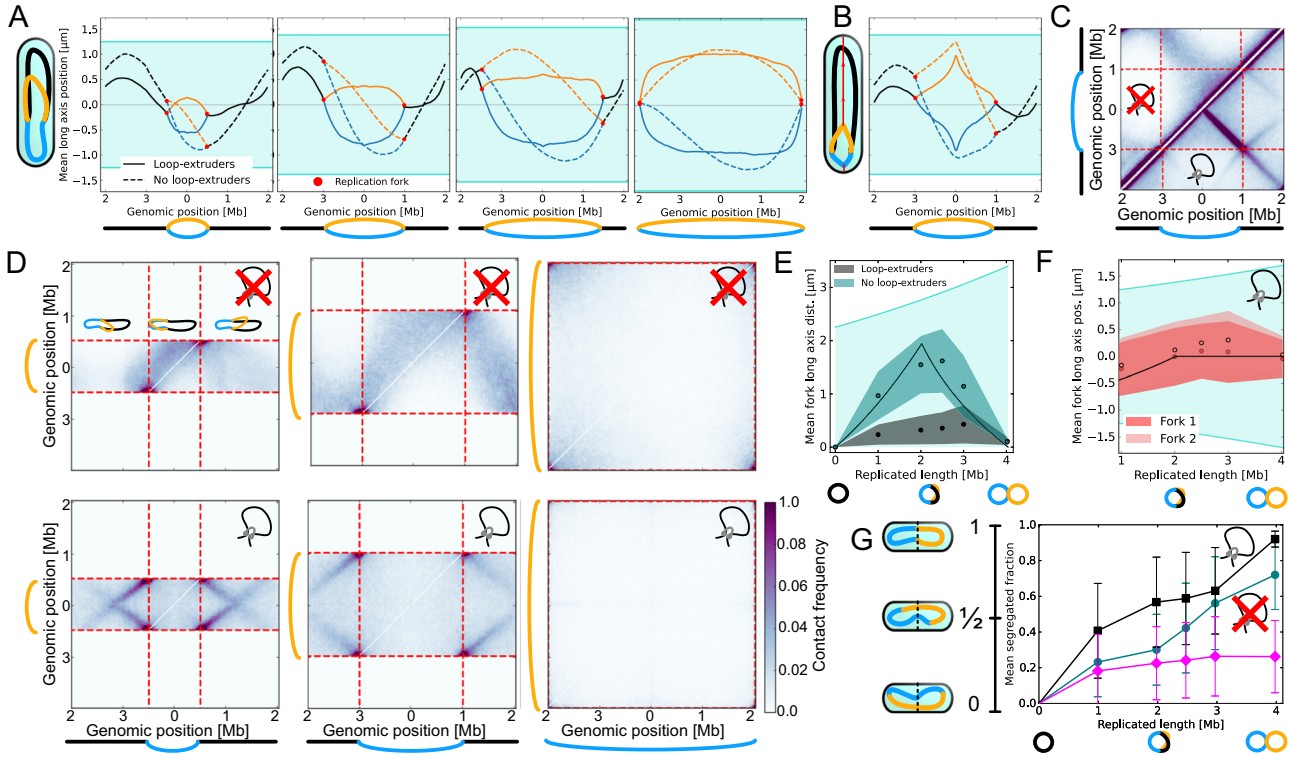

**Fig. 2 | Steady-state simulations of chromosome organization at different replication stages.** Icons with (crossed) loops indicate whether simulations were conducted with (without) loop-extruders. At least four independent steady-state simulations were conducted for each set of simulation parameters (Supplementary Table 2), and statistics were averaged over at least 500 time-points after simulation convergence. **A** Steady-state mean long-axis positions loci as a function of genomic position for $R = 100, 200, 300$, or $402$. Total chromosome length $N = 404$. Standard errors on the mean are narrower than the lines. **B** Steady-state mean long axis positions with *ori*-pulling forces. Data for $R = 200$, Supplementary Fig. 13 shows other replication stages. **C** Intrachromosomal contact maps at $R = 200$. Without loop-extruders (top-left), the chromosome folds from mid-arm, consistent with a left-*ori*-right configuration. With loop-extruders (bottom-right), the chromosome folds from the *ori*, consistent with an *ori-ter* organization. Weaker lines from the replication forks arise from the unreplicated segment

folding towards the *ori*. Color bar in Subfigure D. **D** Inter-chromosomal contact maps for $R = 100, 200, 402$. Without loop-extruders (top row), inter-chromosomal contacts only diminish when $R \approx N$. With loop-extrusion (bottom row), inter-chromosomal contacts diminish already earlier. Insets in top-left map illustrate configurations that would contribute to each section of the map. **E** Mean steady-state long-axis separation between replication forks with or without loop-extruders. Black line gives prediction calculated by assuming that the distance between the forks scales with the shortest genomic length between them (Methods). Turquoise area indicates the confinement length. Ribbons indicate standard deviation. **F** Mean steady-state long-axis position of replication forks with loop-extruders. The distance between *ori* 1 and the replication forks is predicted to scale with $R/2$ for $R < N/2$, after which the forks should stay at mid-cell. Ribbons indicate standard deviation. **G** Segregated fractions as a function of replicated length. Error bars indicate standard deviation.

*ori-ori-ter* ($R < N/2$) or *ori-ter-ori* ($R > N/2$) configurations, as predicted by our segregation model. Furthermore, alignment of the chromosomal arms by loop-extruders implies that the replication forks coincide at roughly the same mean long-axis coordinate, consistent with experimental evidence that replication forks often colocalize in *C. crescentus*[37,58] and other bacterial species[59].

The preferred chromosome orientations can also be seen from simulated contact maps, which measure how frequently two genomic regions are spatially close to each other. The cis-contact maps of chromosomes (Fig. 2C, Supplementary Fig. 6) without loop-extrusion show an off-diagonal line of enhanced contacts starting from mid-arm of the chromosome, indicative of a left-*ori*-right organization. With loop-extruders, by contrast, the off-diagonal line starts from the origin of replication, indicative of an *ori-ter* configuration. We also find that trans-contact maps show less contacts between replicated chromosomal strands with loop-extruders than without (Fig. 2D, Supplementary Fig. 6), further illustrating how loop-extrusion enhances segregation.

To better quantify fork segregation, we analyze the mean long-axis separation between replication forks with or without loop-extruders (Fig. 2E). In the absence of loop-extruders, we observe significant separations between the replication forks, indicative of fork-segregated states. The distance between the forks initially grows, reaches a maximum around half-way through replication, and then decreases. This trend can be qualitatively predicted by presuming that the polymer strands are all equally compressed along the long-axis of the cell (Methods). In the presence of loop-extruders, by contrast, the separation between the replication forks remains small throughout the replication cycle, as expected when the chromosomal arms are zipped together. As replication proceeds, both replication forks move towards mid-cell, as expected for segregated *ori-ter-ori* configurations (Fig. 2F). Our findings indicate that, as replication progresses in the absence of loop-extruders, the forks approach each other as the unreplicated segment between them shrinks, so that there is a smooth transition from fork-segregated to chromosome-segregated states. With loop-extruders, on the other hand, we expect a smooth transition towards segregated states half-way through replication.

Since loop-extruders effectively bring the replication forks together (Fig. 2E), one can ask whether connecting the replication forks into a "replication factory"[5,42,59] would be sufficient to explain the enhancement of segregation we observe due to loop-extrusion. To test this hypothesis, we simulate steady-state configurations of a model where the replication forks are tied together in the absence of loop-extruders. We find that although the replication factory model shows better segregation than the model with neither loop-extruders nor a replication factory, segregation is still significantly better with only loop-extruders (Supplementary Note 7, Supplementary Fig. 7). Even if the replication forks are tied together, in the absence of loop-extruders, the two arms of a newly replicated strand can spread in opposite directions across the long axis of the cell. We hence find that the effective linearization of the origin proximal regions by loop-extruders, which prevents this arm spreading, is important for redirecting entropic forces towards segregation.

To quantify the long-axis segregation of the chromosomes using a simple metric, we calculate the segregated fraction: the fraction of replicated monomers in the correct cell half, minus the fraction in the incorrect cell half. For all simulated values of $R$, we find that the the segregated fraction at steady-state is larger in the presence of loop-extruders (Fig. 2G). To see whether purely entropic forces still lead to some segregation in the absence of loop-extruders, we compare the segregated fraction to values from simulations with an ideal polymer, where chains can freely mix because of the absence of excluded volume interactions (Fig. 2G). For $R \le N/2$, the segregated fraction without loop-extruders is close to that of the ideal polymer, suggesting that purely entropic forces do not lead to significant segregation.

Indeed, as predicted by our segregation model, it is only when we approach the limit of fully replicated chromosomes that entropic segregation without loop-extruders becomes significant, as reflected by the increase in the segregated fraction beyond the ideal polymer reference line. However, even in the limit of fully replicated chromosomes, the segregated fraction remains substantially higher with loop-extruders (mean 0.92, standard deviation 0.02) than without (mean 0.71, standard deviation 0.20). This additional enhancement of segregation at final replication stages appears to be caused by repulsion between chromosomal loops[15] due to off-target loop-extruder loading: when we increase the loading specificity, the segregated fractions when $R \approx N$ are similar with or without loop-extruders (Supplementary Note 8, Supplementary Fig. 9), as we would expect based on our topo-entropic segregation model. Importantly, however, targeted loop-extruder loading is sufficient to give segregation at earlier replication stages. We conclude that although purely entropic forces drive segregation of fully replicated chromosomes, as previously predicted[5], specifically loaded loop-extruders drive segregation at earlier replication stages.

Together, these numerical results confirm our topo-entropic segregation model, showing that purely entropic forces push chromosomes towards fork-segregated left-*ori*-right configurations until late replication stages, and hence inhibit segregation. This is consistent with previous steady-state simulations where half-replicated chromosomes did not demix unless Macrodomain constraints were included[20]. We find that topo-entropic forces directed by loop-extruders can drive chromosome segregation at earlier replication stages. Even when replication is nearly complete and purely entropic forces become segregative, topo-entropic forces still drive stronger segregation. These results suggest that loop-extrusion could significantly enhance segregation during a dynamic replication process.

## Loop-extrusion drives robust simultaneous segregation and replication

Our steady-state simulations at fixed replication stages indicate the direction of (topo-)entropic forces on partially replicated chromosomes. However, it is unclear how these entropic forces affect the dynamics of rapidly replicating chromosomes. Next, we investigate whether chromosomes undergoing dynamic replication in growing cells, where the polymer does not have time to fully relax, will display similar global organization as in the converged steady-state simulations. Importantly, the time-scale of polymer relaxation in our dynamic simulations is set such that the diffusivity of the origin of replication matches experimental data[60,61] (Supplementary Note 9, Supplementary Fig. 10).

All simulations are initialized with a single chromosome in an *ori-ter* orientation. Animations from our simulations readily show that simulated replicating chromosomes do not segregate without loop-extruders (Fig. 3A, Supplementary Movie 1). This contrasts previous work[5] where segregation during replication was achieved using the concentric-shell model. Our simulations with loop-extruders, on the other hand, show clear transitions to segregated states roughly half-way through replication (Fig. 3B, Supplementary Movie 2). This difference is striking: we find that the segregated fraction in the absence of loop-extruders is worse than that of an ideal chain for the first half of replication, and plateaus at roughly 0.2, close to ideal chain values. By contrast, with loop-extruders the segregated fraction reaches values of 0.7 (Fig. 3C). This indicates that, while purely entropic forces in dynamically replicating systems hardly contribute to the dynamic segregation of replicating chromosomes, topo-entropic forces enable rapid large-scale chromosome segregation.

Despite starting from an *ori-ter* configuration before replication, chromosomes without loop-extruders start to rotate towards fork-segregated states during dynamic simulations, as can be confirmed from the average long-axis positions of monomers (Fig. 3D). With loop-

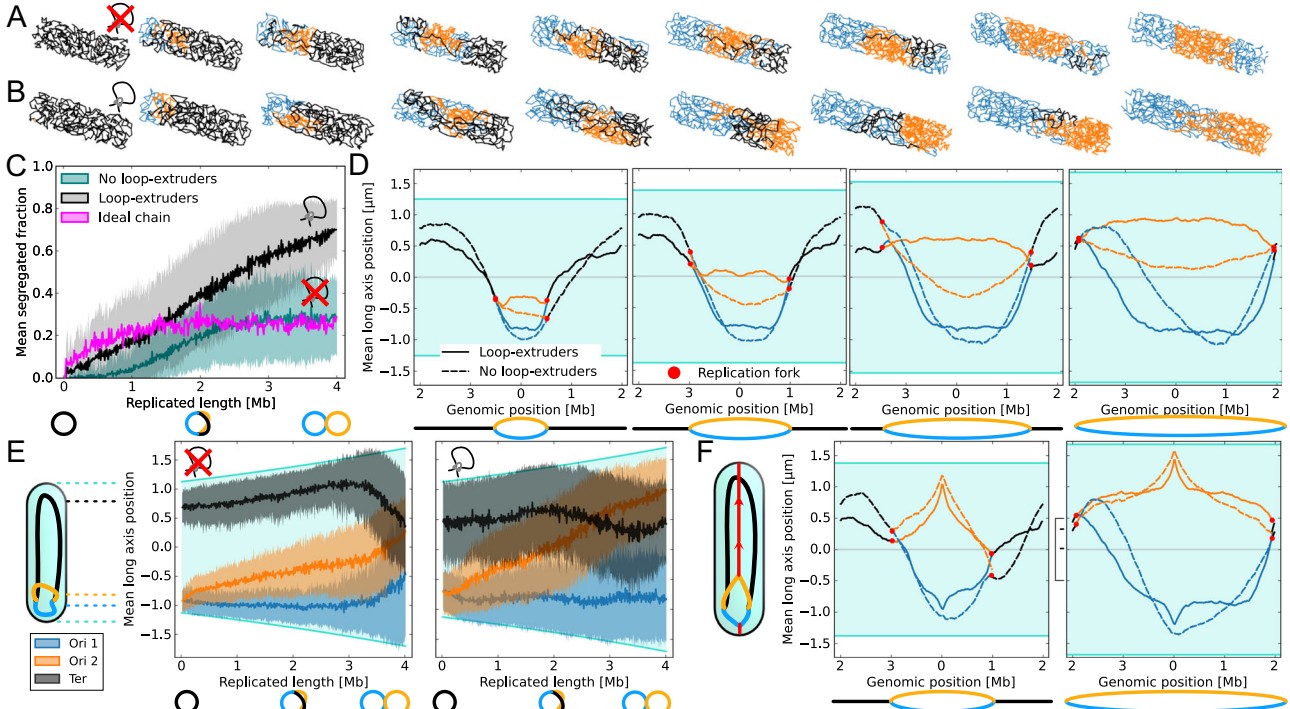

**Fig. 3 | Dynamic simulations of chromosome organization.** At least 100 independent dynamic simulations were conducted for each set of simulation parameters (Supplementary Table 3). For each time-point, statistics were averaged over conducted simulations. **A** Snapshots from simulations of a replicating chromosome without loop-extruders or *ori*-pulling forces, with replication progressing from left to right. Orange and blue lines are newly replicated chromosomal strands, the black line is the unreplicated strand. **B** Snapshots from simulations with loop-extruders but no *ori*-pulling forces. Roughly half-way through replication, the replicated strands start to segregate along the long-axis of the cell. **C** The segregated fraction as a function of replicated length during replication, from simulations with loop-extruders, without loop-extruders, and for an ideal chain without loop-extruders. For clarity, the standard deviation is only shown for simulations with excluded volume interactions, see Supplementary Fig. 8 for the standard deviation of the ideal chain simulations. **D** Mean long-axis positions of chromosomal regions as a function of genomic position from dynamic simulation at different replication stages (as in Fig. 2B). Despite starting from an *ori-ter* organization, without loop-extruders dynamic simulations exhibit fork segregation. Regions around the replication forks appear intermingled. With loop-extruders, segregation is more robust. **E** Mean and standard deviation of long-axis position of genomic regions *ori* 1, *ori* 2, and *ter* across the replication cycle. **F** Mean long-axis positions of chromosomal regions at $R = N/2$ and $R = N$, from simulations with origin-pulling forces. Fork segregation without loop-extruders is still visible for $R = N/2$. At the end of replication, terminal regions are better segregated with loop-extruders.

extruders, on the other hand, the chromosomes maintain clear *ori-ter* configurations throughout the replication cycle. This suggests that loop-extrusion can allow replicating chromosomes to maintain their orientation.

Tracking the *ori*s and the *ter* over time reveals that, without loop-extruders, both origins of replication slowly move towards mid-cell, as expected for left-*ori*-right configurations (Fig. 3E). By contrast, in the presence of loop-extruders, the origins of replication separate at a steady rate as the genomic distance between them grows. The terminal region starts to move towards mid-cell when $R ≈ N/2$, corresponding to the transition from an *ori-ori-ter* configuration to an *ori-ter-ori* configuration predicted by our segregation model.

Overall, our results show that even if replication proceeds rapidly compared to the relaxation of the chromosome, topo-entropic forces directed by loop-extruders drive effective chromosome segregation concurrent with replication, in line with our topo-entropic segregation model. To study how sensitively this result depends on our choice of parameters, we perform simulations with different levels of off-target loading (Supplementary Note 8, Supplementary Fig. 9), different numbers of loop-extruders (Supplementary Note 10, Supplementary Fig. 11) as well as faster loop-extruder off-loading (Supplementary Note 11, Supplementary Fig. 12). Faster off-loading implies that loop-extruders remain localized to narrower region around the origin, as seen in *C. crescentus*[49]. Remarkably, we find that our central results are robust to varying these parameters over a broad range, indicating that specifically-loaded loop-extruders can contribute to bacterial

chromosome segregation even in smaller numbers ( ≳ 20), and even if they do not travel all the way from the origin to the terminus.

## Loop-extruders complement origin segregation by separating terminal regions

To study how loop-extrusion can interact with a ParAB*S*-like mechanism, we include origin-pulling forces in our simulations. We find that in steady-state simulations without loop-extruders, fork segregation occurs even if *ori*-proximal regions are pulled to opposite poles of the cell (Fig. 2B, Supplementary Fig. 13). This result can be rationalized by noting that even if the origins of replication are tethered, fork segregation creates regions where fewer chromosomal strands overlap (Fig. 1A, B). Inclusion of loop-extruders prevents fork segregation, and hence enhances sister chromosome segregation near the replication forks.

In dynamically replicating simulations, we find that early on in the replication process, pulling origins apart leads to faster segregation without loop-extruders, at a rate comparable to that of an ideal chain (Supplementary Fig. 13; Supplementary Movies S3, S4). Since loop-extruders compact and sometimes interlink the replicated strands, loop-extruders can also oppose the stretching of replicated chromosomal strands across the length of the cell at these early replication stages. However, at later replication stages, when entropy favors fork segregation via extension of the unreplicated strand, loop-extruders start to enhance segregation. This is also visible in the mean long-axis positions of the chromosomal regions (Fig. 3F): although most

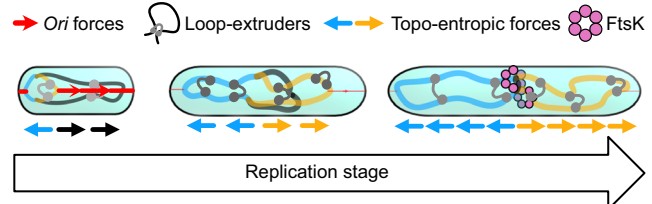

**Fig. 4 | Schematic model of the relative contributions of different bacterial chromosome segregation mechanisms.** At initial replication states, *ori* segregation enables fast separation of newly replicated regions. Around half-way through replication, topo-entropic forces give the most important contribution. Finally, near the end stages of replication, purely entropic forces start to drive chromosome segregation, and factors such as loop-extrusion and FtsK help resolve terminal regions.

genomic regions are clearly segregated to opposite cell halves, *ter*-proximal regions are intermingled without loop-extrusion. We hence find that ParABS-like forces alone can be sufficient to maintain chromosome orientation and to largely segregate bacterial chromosomes as may be expected, but they can be insufficient to efficiently segregate terminal regions. Interestingly, in the presence of loop-extruders, segregation is as accurate with or without origin-pulling in our simulations, but we find that our model with both loop-extrusion and *ori*-pulling is most consistent with experimental Hi-C data from replicating *C. crescentus* cells[36] (Supplementary Note 12, Supplementary Fig. 14).

## Discussion

In this work, we have developed a topo-entropic segregation model, which we confirmed with detailed simulations, to conceptually understand how distinct bacterial chromosome segregation mechanisms work in unison, when during the replication process they are effective, and whether they allow for global or local segregation (Fig. 4). Our model reveals that purely entropic forces cause partially replicated circular chromosomes to adopt unsegregated left-*ori*-right configurations. In accordance, our steady-state simulations without loop-extrusion did not exhibit segregation except at late replication stages, and our dynamic simulations revealed that purely entropic forces hardly contribute to chromosome segregation. Our findings contrast previous simulation results, where a concentric-shell model[5] or constraints on the replication forks[42] were used to achieve concurrent replication and segregation. Unlike these previous models, our work explains why dedicated mechanisms such as loop-extrusion or the ParABS system are necessary to achieve concurrent chromosome replication and segregation: they are needed to overcome segregation-inhibiting entropic forces during replication. Finally, our topo-entropic segregation model elucidates how loop-extruders loaded at the origins of replication physically enhance segregation: the effective linearization of the origin-proximal regions turns segregation-inhibiting entropic forces into segregation-driving topo-entropic forces. Our simulations indicate that loop-extruders loaded at the origin can be sufficient to robustly drive global chromosome segregation by altering the direction of entropic forces: the thermodynamically preferred steady-states in the presence of loop-extruders are segregated, and in dynamic simulations, loop-extruders consequently enable concurrent replication and segregation.

Our simulation results further indicate that, whereas locally acting origin segregation forces are a good segregation mechanism at early replication stages, they do not lead to effective segregation of the terminal regions without loop-extruders. This suggests that ParABS(-like) systems need to be complemented by other mechanisms, such as SMC-mediated topo-entropic forces, repulsion between SMC-loops[15], or terminus segregation mechanisms like FtsK. Thus our model can explain why simultaneously removing condensins and FtsK was found to be lethal in *B. subtilis*[41]. Additionally, at late replication stages even

purely entropic segregation forces become effective, especially if segregation has already been initiated[62,63], suggesting that in species such as *C. crescentus*, origin segregation could be responsible for initiation, and entropic forces together with FtsK (essential in *C. crescentus*[26]) for completion of segregation. Loop-extrusion directed topo-entropic forces, by contrast, appear to facilitate chromosome segregation globally, and can be efficient even without origin segregation mechanisms, as observed in ParA-lacking mutants of *B. subtilis*[64].

Our work gives additional experimentally testable predictions: first, in the absence of loop-extruders loaded at the origin, replication fork separation should be observable; second, as the "ends" of effective linear segments, loop-extruder loading sites should be the first loci to segregate after replication; and third, in systems with sufficiently efficient loop-extrusion and an origin segregation system, loop-extrusion might be sufficient for chromosome segregation, whereas locally acting origin segregation forces should always be complemented by a terminus-segregation mechanism. Further work could also explore whether entropic fork segregation plays a role in establishing left-*ori*-right chromosome states in some bacteria, such as in E. coli[65] and transiently in *B. subtilis*[66].

Our work also provides insight into possible mechanisms of loop-extrusion in bacteria. In our model, we employed nontopological loop-extruders capable of by-passing obstacles[51,67,68], as opposed to topological loop-extruders that trap a chromosomal strand in a ring[69]. When we repeated our simulations with topological loop-extruders, we found that chromosomal segregation was drastically inhibited (Supplementary Note 13, Supplementary Fig. 15). This suggests one possible reason why nontopological loop-extrusion might be preferred by bacteria.

The physical principles we have elucidated—on how entropy, specifically loaded loop-extruders, and locally acting forces act together to segregate and organize bacterial chromosomes—could be used to design segregation mechanisms for synthetic cells[70] or "genomes-in-a-box"[71]. Finally, although our work has focused on SMCs loaded at the origins of replication on an unstructured circular chromosome, our model for how topo-entropic forces orient polymers could be adapted to include structures such as Macrodomains[20] or CIDs[36], or for prokaryotic or eukaryotic systems with different patterns of loop-extruder loading.

## Methods

### Summary of simulation methods

We built on previously published code for loop-extruders capable of by-passing each other on DNA[51], and extended this code for replicating chromosomes. This simulation scheme couples 1D simulations of loop-extruders to 3D simulations of a bead-spring polymer. Supplementary Notes 3-6 contain more detailed information.

We keep 1D loop-extruder dynamics the same as in the original simulations: loop-extruders are preferentially loaded at the origins of replication, they have a constant unbinding rate that is increased near the terminus, and during collisions between two loop-extruders, the loop-extruders either stall, unbind, or bypass each other.

To model replication in the 1D simulations, we introduce two replication forks that proceed independently from *ori* to *ter*. As the forks proceed, loop-extruders can move and be loaded onto the newly replicated sites. To keep the density of loop-extruders constant, we increase the number of loop-extruders as replication proceeds. When a loop-extruder and a replication fork collide, the loop-extruder either unbinds or jumps to either strand on the other side of the replication fork.

For replicating simulations, we run a 1D simulation until replication is complete, and save the loop-extruder and replication fork positions, so that these saved positions can then be used in 3D simulations.

We initialize replicating 3D simulations with two bead-spring polymer rings, one of which corresponds to the unreplicated chromosome. Excluded volume interactions are modeled by a repulsive polynomial potential[51]:

$$U(r) = U_0 \left( 1 + \left( \frac{r}{R_{max}} \right)^{12} \left( \frac{6}{7} \left( \frac{r}{R_{max}} \right)^2 - 1 \right) \right),$$

where $U_0$ is the depth of the potential (1.5 $k_B$T for most of our simulations); $r$ is the separation between two beads; and $R_{max}$ is the distance at which the potential's derivative becomes zero, 1.05 times the spring length in our simulations. This potential allows some strand passing, and our simulations hence do not conserve the topology of the replicating chromosome.

Unreplicated monomers are bound by springs to their replicated copies, but the unreplicated monomers have no excluded volume interactions. The monomers are constrained to a cylindrical confinement using a harmonic potential, and additionally, we can introduce forces that pull the origins to specific positions along the long axis of the confinement.

At each simulation time-step, we fetch the positions of the loop-extruders and replication forks from the saved 1D simulation trajectories. Harmonic bonds bind monomers connected by a loop-extruder. When a replication forks moves, we turn on the excluded volume interaction for the newly replicated monomer, and turn off the potential that tied it to its replicate. We increase the length of the cylindrical confinement to model cell growth. If origin-pulling forces are used, the positions the *oris* are constrained to are also updated. Given these updated constraints, the 3D polymer is given some time to relax, and the 3D polymer configuration is then saved.

Steady-state simulations are run similarly. We allow loop-extruder movement, but fix the genomic locations of the replication forks. Additionally, the confinement length and possible origin-pulling forces are kept constant.

### Orienting sampled chromosome configurations

Since the two replicated strands and two arms of a partially replicated chromosome are indistinguishable, when averaging over many simulations, the mean positions of both replication forks and both *oris* are equal. Hence, to distinguish whether chromosomes or replication forks segregate, each simulated configuration is oriented to break these symmetries. We define Strand 1 such that the center of mass of Strand 1 and the unreplicated segment is closer to mid-cell than Strand 2 and the unreplicated segment (Supplementary Fig. 16A). Pole 1 is defined as the pole closer to Strand 1. Arm 1 is defined as the arm of Strand 1 and the unreplicated segment that is closer to Pole 1 (Supplementary Fig. 16B). When *ori*-pulling forces are simulated, the strands are not renamed, since the applied forces distinguish the two replicated strands.

In replicating simulations, the direction of replication distinguishes the terminal region. In steady-state simulations with loop-extruders, the terminal region lacks a loop-extruder loading site. However, in steady-state simulations without loop-extrusion, when $R = N/2$, all three chromosomal strands are indistinguishable. In this case, we label Strand 1 as the one with a center of mass closest to a pole, Strand 2 as the strand closest to mid-cell, and Strand 3 as the strand closest to the other pole.

### Predicting fork separation or position

To predict the steady-state fork separation in the absence of loop-extruders, we suppose that the long-axis extension per monomer is constant, and that the mean separation between monomers at opposite poles is $L_{max}$. We use a maximum distance $L_{max} = L - d$, since each monomer is on average in the middle of a confinement blob of radius $d/2$[46], and hence on average a distance $d/2$ from the cell pole. We note that this is a simple approximation; a proper analysis should account for overhangs, where one strand extends beyond the region of overlap[62], and the effect of boundaries on the monomer density.

For the first half of the replication cycle ($R < N/2$) the extension per monomer is $L_{max}2/N$, since the largest ring of total length $N$ stretches from pole to pole and back. The expected distance between the forks, separated by a linear strand of length $R$, is hence $L_{max}2R/N$.

For the second half of the replication cycle ($R > N/2$), the largest ring has length $2R$, so that the extension per monomer is $L_{max}/R$. This means that the forks, separated by a linear segment of length $N - R$, should now have a mean distance $L_{max}(N - R)/R$. Combining these two expressions gives the black line in Fig. 2E.

Similarly, in the presence of loop-extruders, we expect that for $R < N/2$, the *ori* and the replication forks are separated by a linear segment of length $R/2$, and their expected separation is hence $L_{max}R/N$. Once $R \geq N/2$, the chromosome adopts an *ori-ter-ori* configuration, with the replication forks located mid-cell. This gives the black line in Fig. 2F.

### Reporting summary

Further information on research design is available in the Nature Portfolio Reporting Summary linked to this article.

## Data availability

The simulation data generated in this study have been deposited to Zenodo under the https://doi.org/10.5281/zenodo.10877077.

## Code availability

The Python 3.0 code used for simulations in this study can be found on GitHub[72]. It uses the open-source packages Polychrom 0.1.1[51] and OpenMM 7.6.0.[73]. The Julia 1.8.5. code used for analysis in this study can be found on GitHub[74].

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

## Acknowledgements

We thank Cees Dekker, Bela Mulder, Valerio Sorichetti, and Tom Brandstätter for discussions, as well as Hugo Brandão and Leonid Mirny both for discussions and help with simulations.

## Author contributions

J.H. and C.P.B. designed the study. J.H. and M.C.F.v.T. conducted research. J.H. performed simulations and conducted analysis. J.H. and C.P.B. wrote the manuscript with input from M.C.F.v.T. C.P.B. supervised the work.

## Competing interests

The authors declare no competing interests.
