## [Peer Review File · Nature Communications]

Loop-extruders alter bacterial chromosome topology to direct entropic forces for segregationEditorial Note: Parts of this Peer Review File have been redacted as indicated to remove third-party material where no permission to publish could be obtained.

REVIEWER COMMENTS

Reviewer #1 (Remarks to the Author):

In this manuscript titled "Loop-extruders alter bacterial chromosome topology to direct entropic forces for segregation" the authors suggest a model that describes the driving force for chromosome segregation in bacterial cells. This question of how two copies of the chromosome manage to fully segregate prior to cell division is a long-standing open question, with high importance to the field of bacterial cell cycle. A prior work has suggested that two chromosomes confined in a cell will segregate purely due to entropic principles. In this work, the authors challenge this prior model and elegantly show that in the dynamic case of a replicating chromosome entropic forces are not enough to drive chromosome segregation. Furthermore, the authors show that the addition of loop extruders to the model does result in chromosome segregation also in the dynamic case of a replicating chromosome. These theoretical results correspond well with known experimental results based on genetic perturbations in the bacteria *B. subtilis* and *C. crescentus*.

The authors base their claims first on a biophysical model, and second on a molecular dynamics simulation of a replicating chromosome. The description of the simulation part is detailed and clear and the authors justify further their choice of simulation parameters in the supplementary notes. However, the description of the biophysical model is very heuristic, and in my opinion, would benefit from a clearer grounding of the presented results.

I think that this work provides an important theoretical contribution to the ongoing discussion about the driving forces for chromosome segregation in bacteria. The model provides several predictions that can be experimentally tested, and I think its publication can benefit a wide community of both theoretical and experimental researchers in the fields of chromosome dynamics and bacterial cell cycle.

A few detailed comments for the authors:

1. In the sections that describe the biophysical model (sections titled "Entropy does not segregate partially replicated circular chromosomes." and "Loop-extrusion redirects entropic forces enabling segregation during replication."), I would like to see a clear justification for the conclusions presented in these sections. While the explanations of what are the entropically preferred orientations make sense intuitively, I would like to see a more theoretical grounding for these results. I admit that I don't have a theoretical background in polymer physics, and therefore I don't understand the reasoning for some results that might be trivial for people in this field, but since this journal addresses a wide audience, I think this section would benefit from clearer reasoning. If these results are based on established theoretical results from the field of polymer physics, then the authors should refer to these results, otherwise, the authors should provide a more detailed model to show that their claims are supported by this model (I acknowledge that the claims are well supported by the simulations presented later in the manuscript, but this comment relate to the biophysical model).

2. The authors do provide a more detailed model in supplementary notes 1 & 2, but according to the main text, this model aims to address a few simplifications that were taken in the model presented in the main text. Hence, I still wish to see some more support for the model presented in the main text. As for the blob analysis presented in the supplementary note, I think this model description can be better clarified for a wider audience, who are not familiar with polymer physics theory. For example, the authors write "This immediately shows that it is entropically costly to have more polymer strands in parallel along the tube, since the average blob size is smaller". In my opinion, for researchers with no background in polymer physics, this claim is not immediately seen.

3. Figures 1F and 1G are a bit unclear. I struggled to understand the meaning of the different arrows and markings. Eventually after carefully reading the legend I managed to understand the figures, but I think they can be improved by adding short text labels on the figure itself.

4. On page 5 there is a reference to (Supplementary Note 8, Supplementary Fig. 11), I believe it should be (Supplementary Note 8, Supplementary Fig. 7) instead.

Reviewer #2 (Remarks to the Author):

The work by Harju et al uses theory and simulation to investigate genome segregation in bacteria. Using a polymer model the manuscript shows that addition of loop-extruders that bind to the genome and constrain its topology enhances the entropic forces for genome segregation. The authors claim that this model explains the necessary role of loop extruders and the mechanism of genomic separation in bacteria. The manuscript is generally well written, but would benefit from clarification regarding assumptions in the model, the topology constraints and kinetics, and their effects on the results. I can well judge theory/simulations but am not an expert on the biology of genome separation. I think the manuscript is suitable for Nat. Comm., but a few clarifications regarding model and conclusions are required. Moreover, quantitative comparison to existing experimental data would strengthen the claims presented.

- This is a theory/simulation paper, yet there is hardly any information on the model used and associated assumptions explained in the main text. In the introduction authors claim that their model is better than the previous "concentric-shell model" used that already demonstrated entropic forces in genome segregation, but it's not clearly explained why. Given that the new model predicts qualitatively different results from the old one; no segregation without loop extruders vs. segregation, a clear explanation is warranted. I also strongly suggest putting the relevant theory/simulation setup details in the main text, e.g., in the method section.

- The manuscript appears to be written for a more specialised audience that knows what loop-extruders are, what ori-ori-ter genomic configurations are and how are they different from ori-ter-ori. For the wide readership of Nat. Comm. I suggest adding an intro Figure that clearly explains what loop extruders are, how the function and the jargon used to describe genome configurations, or at least very clearly describe this in the text.

- Page 3, the MS claims that topo-entropic segregation model was developed. Please clarify what topological constraints were included in the model. As far as I understood, a blob polymer model was used. Can the genome pass through itself or is the topology preserved? Can loop extruders pass through each other or not?

- Page 3: Kinetics needs to be discussed. The MS shows that entropic forces with loop-extruders help at intermediate stages of replication. However, this is based on purely equilibrium arguments. If the genome can fully relax, then thermodynamic forces at intermediate stages are presumably largely irrelevant. On the other hand, if genome cannot relax on replication timescales, how relevant are the equilibrium entropic arguments? Can the genome, a very long polymer, reach equilibrium configurations on the replication timescales? What is the role of possible entanglements? A quantitative comparison of genome replication vs. segregation kinetics is necessary to show whether entropic arguments are relevant at intermediate replication stages.

- Is it possible that the main benefit of loop extruders in segregation is to guide kinetics, prevent traps and entanglements during genome replication, rather than slight modification to equilibrium entropic forces? Please discuss.

-

- Fig 2a,b. Is this a result of a single simulation? No error bars are shown. How replicable are these results? Please add error bars, or at least somehow indicate what is the statistical significance between the two curves.

- Fig 3C, why does the segregated fraction initially decrease without loop extruders?

- Discussion section: Here authors state: "dynamic simulations reveal that purely entropic forces hardly contribute". If this is the case, why spend the first half of the manuscript discussing entropic forces?

Clarify whether the main role of loop extruders is to modify kinetics or thermodynamic forces for segregation.

- Later in the discussion section states: "in our model, we employed non-topological loop extruders" The model is called topo-entropic and the title claims topology is considered. Related to a point raised above, clarify exactly which genome/loop-extruder topologic constraints are taken into account in the model and which are not.

- Direct quantitative comparison to existing experimental data (if they exist) would strengthen the claims presented.

Reviewer #3 (Remarks to the Author):

Harju et al. report on their theoretical analysis and modeling of bacterial chromosome segregation to resolve the roles of three factors that have been proposed to assist in or drive segregation: chromosome configurational entropy, loop-extruding SMC complexes, and dedicated/directed forcing mechanisms, such as the ParABS system. In contrast to previous models, the authors consider all of these mechanisms together and their possible complementary roles, as well as the complication of chromosome replication while segregation proceeds. They find that although entropy of ring polymer chromosomes can in principle lead to segregation, this is not the ideal kinetically because segregation would occur more slowly than replication, and moreover, segregation is not the favored equilibrium state until after replication has been completed. Loop extrusion that 'zips' together chromosomal arms, given the model proposed by the authors, is sufficient to rescue an entropic mechanism and leads to more robust segregation outcomes. Similarly, ParABS-like mechanisms can segregate sister oris, but without chromosome compaction/'linearization' via loop extrusion, they are deficient. The authors argue that their model can explain the importance of origin-loaded loop extruders and the notable "secondary diagonal" observed in Hi-C experiments in bacteria.

Altogether, this is a very well considered effort to understand seemingly disparate mechanisms/factors of chromosome segregation. The work addresses valuable questions about the importance and interplay of these mechanisms, and uses simple, but fundamental and largely convincing physical arguments to do so. These simple theoretical arguments are then supported by extensive simulation work. From a technical standpoint, the authors excelled in developing simulations that could properly interface the various factors they considered. I also appreciated the model of the replicating chromosome, and it is the first of its kind of which I am aware. However, in places the authors were extremely terse where additional explanation would be helpful. Recognizing, that word count may be the issue, I generally think the manuscript reads well within these constraints, but there may be room for improvement. I also have a few questions about the choice of extrusion model in the context of replication given recent results suggesting that even non-topologically bound SMC complexes can be blocked by replication and transcription machinery. Altogether, this largely well written and well considered manuscript provides a good conceptual advance for the fields of chromosome biophysics and bacterial biology/biophysics.

Major comments:

1. Figure 2 and the main text describing it is difficult to follow.

a) It might be helpful to readers to add markers for replication forks to plots like Fig 2A or at least

directly stating that the fork is at the convergence of the orange, blue, and black lines.

b) The authors may want to more clearly explain why different metrics are used to evaluate different scenarios in Fig 2E-F.

c) The colors, labeling of colors, and shading in Fig 2E-F are also confusing. Confinement length is gray, but why is it a region not a line? Turquoise is estimated maximum separation — is that the dark gray or the light blue? and why are these regions instead of lines?

d) What does Fig 2G look like for the replication factory model?

2. It is argued that loop extrusion “linearizes” the chromosome arms.

a) the meaning of this statement is not fully explained in the main text. It becomes clearer when the idea is revisited late in the manuscript or upon a close look at the supplemental scaling argument, but a few additional words here would help readers.

b) what is the threshold for having “enough” loop extruders to “linearize” the chain? given ~40 SMCs (or maybe just a few!), it’s not obvious that the ‘zipped’ chromosome isn’t more like a few connected rings rather than a linearized chain.

c) It is hypothesized that the “improvement of segregation by loop-extruders could be due to repulsion between chromosomal loops caused by off-target loop-extruder loading”. I thought it was argued that “linearization” by extruders is responsible for segregation improvements. Furthermore, the hypothesis can be directly tested by the authors by simulating without off-target loading.

3. The authors consider models where extrusion is either: 1) ‘non-topological’ so extrusion bypass replication forks and effectively jump to a new strand or engulf the fork or 2) ‘topological’ so the SMC encloses a replicated strand after encountering a replication fork. They also consider these models in the scenario where fork encounters facilitate SMC unloading.

But several recent papers argue that loop extruders might be pushed or otherwise obstructed by other complexes such as those involved in replication or transcription (ref 42 in this manuscript + Dequeker et al. Nature 2022, Jeppsson et al. Sci Adv 2022, Banigan et al. PNAS 2023). How might this alter the results?

Minor comments:

1. A summarizing sentence or two at the end of the 2nd results subsection “entropy does not segregate...” may improve clarity. As I understand it, the equilibrium ordering of ori and ter arises from maximal elongation of the largest rings, which differs between the first and second halves of replication, but somehow this logic doesn’t come through as clearly as it could.

2. please check supp fig references. e.g., rep factory is Fig S7, not S11 as stated in text fig s7.

3. The color key in Fig. 3C is a little confusing because of the reuse of gray and it being left to the reader to decipher the meaning of black/black outlines.

4. It would be valuable to include a supplemental table of major simulation parameters.

5. Does this work offer any insights about how entropy and other factors could segregate chromosomes in spherical bacteria?

6. In the supplement it is stated that “It cannot be assumed that the free energy cost per blob is the same for single chromosomal strands and doubled-up strands; since the doubled-up strand has more degrees of freedom, its free energy cost per blob is expected to be higher.” This is a confusing since

blobs are usually defined by kT ! I guess it doesn't matter since there are more higher energy blobs, so the argument holds, but the authors might want to clarify.

Reviewer 1 (Remarks to the Author):

In this manuscript titled “Loop-extruders alter bacterial chromosome topology to direct entropic forces for segregation” the authors suggest a model that describes the driving force for chromosome segregation in bacterial cells. This question of how two copies of the chromosome manage to fully segregate prior to cell division is a long-standing open question, with high importance to the field of bacterial cell cycle. A prior work has suggested that two chromosomes confined in a cell will segregate purely due to entropic principles. In this work, the authors challenge this prior model and elegantly show that in the dynamic case of a replicating chromosome entropic forces are not enough to drive chromosome segregation. Furthermore, the authors show that the addition of loop extruders to the model does result in chromosome segregation also in the dynamic case of a replicating chromosome. These theoretical results correspond well with known experimental results based on genetic perturbations in the bacteria *B. subtilis* and *C. crescentus*. The authors base their claims first on a biophysical model, and second on a molecular dynamics simulation of a replicating chromosome. The description of the simulation part is detailed and clear and the authors justify further their choice of simulation parameters in the supplementary notes. However, the description of the biophysical model is very heuristic, and in my opinion, would benefit from a clearer grounding of the presented results. I think that this work provides an important theoretical contribution to the ongoing discussion about the driving forces for chromosome segregation in bacteria. The model provides several predictions that can be experimentally tested, and I think its publication can benefit a wide community of both theoretical and experimental researchers in the fields of chromosome dynamics and bacterial cell cycle.

We thank the referee for their positive and encouraging assessment of our work, as well as for their constructive feedback. We fully agree that the question of how bacteria achieve chromosome segregation is an interesting and long-standing question in the field.

Based on the referee’s detailed feedback, we have made efforts to clarify the theoretical arguments presented in our manuscript. We feel that these edits make our work more accessible to the broad readership of Nature Communications, and we would like to thank the referee for their helpful input.

A few detailed comments for the authors:

1. In the sections that describe the biophysical model (sections titled “Entropy does not segregate partially replicated circular chromosomes.” and “Loop-extrusion redirects entropic forces enabling segregation during replication.”), I would like to see a clear justification for the conclusions presented in these sections. While the explanations of what are the entropically preferred orientations make sense intuitively, I would like to see a more theoretical grounding for these results. I admit that I don’t have a theoretical background in polymer physics, and therefore I don’t understand the reasoning for some results that might be trivial for people in this field, but since this journal addresses a wide audience, I think this section would benefit from clearer reasoning. If these results are based on established theoretical results from the field of polymer physics, then the authors should refer to these results, otherwise, the authors should provide a more detailed model to show that their claims are supported by this model (I acknowledge that the claims are well supported by the simulations presented later in the manuscript, but this comment relate to the biophysical model).

Based on this feedback, we have made an effort to better introduce and explain the relevant concepts from polymer physics in the Results section, and have also included a few new references to provide links to established theoretical results (Ref 48 on page 2).

Our theoretical argument relies on a well-established result from polymer physics: a polymer with excluded volume interactions (referred to in the literature as a ‘real’ polymer, as apposed to an ‘ideal’ polymer) confined to a sufficiently long tube will extend along the tube’s long axis to maximize its entropy. We use this argument several times in our manuscript: first, to argue that just a ring polymer should extend inside the cell. Second, to argue that a linear strand attached at the replication forks should extend parallel to the ring, thereby inhibiting segregation. And, finally, to argue that a chromosome zipped up and linearized by loop-extruders should also extend within the cell. We hope that this explanation, as well as our edits, clarify the theoretical basis of our arguments.

2. The authors do provide a more detailed model in supplementary notes 1 & 2, but according to the main

text, this model aims to address a few simplifications that were taken in the model presented in the main text. Hence, I still wish to see some more support for the model presented in the main text. As for the blob analysis presented in the supplementary note, I think this model description can be better clarified for a wider audience, who are not familiar with polymer physics theory. For example, the authors write “This immediately shows that it is entropically costly to have more polymer strands in parallel along the tube, since the average blob size is smaller”. In my opinion, for researchers with no background in polymer physics, this claim is not immediately seen.

We appreciate these suggestions, which have helped make our manuscript more accessible to readers with different research backgrounds. Supplementary Notes 1 and 2 address the following simplification: in our above argument, we did not consider the fact that when a ring and a linear segment extend in parallel across their confinement, they both have less space along the radial axis of the confinement tube. Supplementary Note 1 addresses this issue by showing that, even if we consider this reduced diameter of confinement for each strand, fork segregated states are still entropically preferred in the absence of loop-extruders.

We have adjusted the language of these Supplementary Notes 1 and 2 to make them more accessible to readers without prior knowledge of polymer blob theory. For instance, we adjusted the sentence quoted by the referee to:

“When more polymer strands lie in parallel along the tube, the average blob size decreases, and the number of blobs per unit length subsequently increases. Since the entropic cost of confinement scales with the number of blobs, this tells us that it is entropically costlier to have more strands in parallel along the length of the tube.”

3. Figures 1F and 1G are a bit unclear. I struggled to understand the meaning of the different arrows and markings. Eventually after carefully reading the legend I managed to understand the figures, but I think they can be improved by adding short text labels on the figure itself.

We realized that we were using arrows somewhat inconsistently in the figure. We removed some arrows, and left only those that indicate loop-extruder dynamics. In addition, as suggested by the referee, we added labels that indicate the replication forks, as well as loop-extruder loading and unbinding. These labels make the figure easier to interpret, and we thank the referee for their suggestions.

4. On page 5 there is a reference to (Supplementary Note 8, Supplementary Fig. 11), I believe it should be (Supplementary Note 8, Supplementary Fig. 7) instead.

Thank you, we have now corrected this in our revised manuscript.

Reviewer 2 (Remarks to the Author):

The work by Harju et al uses theory and simulation to investigate genome segregation in bacteria. Using a polymer model the manuscript shows that addition of loop-extruders that bind to the genome and constrain its topology enhances the entropic forces for genome segregation. The authors claim that this model explains the necessary role of loop extruders and the mechanism of genomic separation in bacteria. The manuscript is generally well written, but would benefit from clarification regarding assumptions in the model, the topology constraints and kinetics, and their effects on the results. I can well judge theory/simulations but am not an expert on the biology of genome separation. I think the manuscript is suitable for Nat. Comm., but a few clarifications regarding model and conclusions are required. Moreover, quantitative comparison to existing experimental data would strengthen the claims presented.

We thank the referee for their time and constructive feedback, as well as for supporting the publication of our work. Below, we respond to the points raised by the referee and offer further clarification on the questions raised.

- This is a theory/simulation paper, yet there is hardly any information on the model used and associated assumptions explained in the main text. In the introduction authors claim that their model is better than the previous “concentric-shell model” used that already demonstrated entropic forces in genome segregation, but it’s not clearly explained why. Given that the new model predicts qualitatively different results from

[redacted]

Figure R1: **Depiction of the concentric-shell model.** Figure 4 from Jun and Mulder (2006).

the old one; no segregation without loop extruders vs. segregation, a clear explanation is warranted. I also strongly suggest putting the relevant theory/simulation setup details in the main text, e.g., in the method section.

We appreciate the referee’s request for more details, and agree that explaining the difference to the concentric-shell model is critical for explaining the relevance of our work.

Jun and Mulder (2006) define a concentric-shell model as follows. At intermediate replication stages, one replicated strand and the unreplicated strand of the circular chromosome are confined to a cylinder of radius R_{in} , whereas the second replicated strand is confined to a cylinder of radius $R_{\text{out}} > R_{\text{in}}$ (Figure R1). This model assumes an inherent asymmetry in accessible volume between the two replicated chromosomal strands. In our work, by contrast, we assume no such asymmetry.

Jun and Mulder (2006) state that this concentric-shell mechanism is necessary to achieve concurrent replication and segregation in their simulations. Qualitatively, their finding is consistent with our dynamic simulations: we also find that *without any additional mechanisms*, replicating chromosomes in our simulations fail to segregate.

The theoretical discussion by Jun and Mulder (2006) focuses on the limit of fully replicated chromosomes, and relies on scaling arguments that show that two ring polymers confined to a sufficiently long tube will entropically segregate. To our knowledge, our work is the first to theoretically consider the direction of entropic forces on a *partially replicated* chromosome. Since chromosome replication is often on-going for most of the bacterial cell cycle, this is a biologically relevant case to study. We find that at these intermediate replication stages, purely entropic forces inhibit rather than enhance bacterial chromosome segregation. Our theory can therefore explain why Jun and Mulder (2006) did not observe concurrent replication and segregation without a concentric-shell mechanism in their simulations.

Finally, our work also discusses how specifically loaded loop-extruders can enable entropic segregation concurrent with replication by effectively linearizing the bacterial chromosome. To our knowledge, this is the first theoretical work that discusses how specifically loaded loop-extruders would impact the direction of entropic forces during bacterial chromosome segregation.

Based on the reviewer’s feedback, we revised our manuscript as follows:

1. We adjusted our introduction to better explain the concentric-shell model as well as the simulation results presented by Jun and Mulder (2006).
2. We rephrased a sentence in the introduction referring to previous simulation work by Junier *et al.* 2014.
3. We edited the theory section to make it more accessible.
4. We included a summary of our simulation approach in the Methods.
5. We moved the section describing our simple prediction for the fork separation from the SI into the Methods.

Finally, we note that we considered moving the blob theory arguments to the Methods section as suggested by the referee, but we felt that it would serve readers better to keep this as a more extensive theoretical discussion next to Supplemental Figure 1 in the SI.

- The manuscript appears to be written for a more specialised audience that knows what loop-extruders are, what ori-ori-ter genomic configurations are and how are they different from ori-ter-ori. For the wide readership of Nat. Comm. I suggest adding an intro Figure that clearly explains what loop extruders are, how the function and the jargon used to describe genome configurations, or at least very clearly describe this in the text.

As the referee points out, Nature Communications has a wide readership, and we fully agree that it is important to make our text accessible to readers with different backgrounds. We have therefore made efforts to better explain all terminology used in our manuscript by introducing the following changes:

1. We added a sentence to explain the concept of loop-extrusion into the introduction.
2. We included a small icon to indicate how loop-extruders form loops to Subfigure 1E.
3. We added labels to Figure 1A,D,E to indicate examples of left-ori-right, ori-ter-ori, and ori-ori-ter configurations.

We feel that these changes clarify our manuscript to readers unfamiliar with this terminology, and thank the referee for their helpful feedback.

- Page 3, the MS claims that topo-entropic segregation model was developed. Please clarify what topological constraints were included in the model. As far as I understood, a blob polymer model was used. Can the genome pass through itself or is the topology preserved? Can loop extruders pass through each other or not?

We apologize for the confusion, and recognize that “topology” can refer to many aspects of chromosome organization.

Our motivation for introducing the word “topo-entropic” to contrast “purely entropic” is that we claim that specifically loaded loop-extruders can contribute to bacterial chromosome segregation is by “zipping up” the arms of the bacterial chromosome, by imposing constraints that *effectively change the large-scale topology of the chromosome from a ring to a linear segment*. Furthermore, we argue that the chromosome topology and geometry during replication (i.e. three strands connected at the forks) matter for the directions of entropic forces.

Our simulations are indeed based on a coarse-grained bead-spring model, where monomers can occasionally pass through each other with a rate set by excluded-volume interactions. As the referee points out, this means that the chromosome’s topology is not truly preserved, and entanglements will be resolved, as occurs in cells due to topoisomerases.

Finally, the referee asks about whether loop-extrusion in our model is topological or not. As stated in our manuscript, our main simulation results are from a model where loop-extruders can bypass each other after a brief stalling period, and the loop-extruders are therefore non-topological. Crucially, these non-topological loop-extruders can still zip up the two arms of the chromosome, resulting in the effective change of chromosome topology from circular to linearized that our model relies on.

Based on the referee’s feedback, we have made an effort to clarify what we mean by the use of the word “topology” on page 3. We have also replaced the phrase “topological constraint” by “constraint” to avoid confusion.

- Page 3: Kinetics needs to be discussed. The MS shows that entropic forces with loop-extruders help at intermediate stages of replication. However, this is based on purely equilibrium arguments. If the genome can fully relax, then thermodynamic forces at intermediate stages are presumably largely irrelevant. On the other hand, if genome cannot relax on replication timescales, how relevant are the equilibrium entropic arguments? Can the genome, a very long polymer, reach equilibrium configurations on the replication timescales? What is the role of possible entanglements? A quantitative comparison of genome replication vs. segregation kinetics is necessary to show whether entropic arguments are relevant at intermediate replication stages.

As the referee correctly points out, our steady-state simulations indicate the entropically preferred chromosome configuration at equilibrium. However, we also performed dynamic simulations of a replicating chromosome, which may not have time to fully relax to its equilibrium configuration during the cell-cycle (Fig. 3 in the manuscript). The time-scale of our replicating simulations was set so that the short-time scale diffusion of the *ori* roughly matched experimental data from live cells. This way, we can model segregation kinetics during replication.

Importantly, these dynamic simulations (see Fig. 3) demonstrate that purely entropic forces without loop-extruders do not contribute to segregation during replication. This challenges the long-standing hypothesis that purely entropic forces could explain bacterial chromosome segregation. In addition, our dynamic simulations show that topo-entropic forces in the presence of loop-extruders are sufficient to drive effective segregation on physiologically relevant time-scales. Thus, our dynamic simulations show that the qualitative difference in segregation that we observe with and without loop-extruders (see Fig. 3) is further emphasized in a dynamically replicating setting.

Although our dynamic simulations indicate that replicating chromosomes in cells don't fully relax to equilibrium during the cell-cycle (especially in the purely entropic case), it is still theoretically important and conceptually insightful to establish what the equilibrium states are, and how the entropic forces that drive the system towards these states are oriented in a dynamic setting.

Entanglements are certainly known to play a role during bacterial chromosome segregation; for instance, the two replicated strands right behind the replication forks are known to wind around each other, and these entanglements must be resolved by topoisomerases. In our coarse-grained simulations, entanglements are resolved via strand-passage with rates based on prior work (Brandão et al., 2019, 2021), but our coarse-grained simulation scheme is not suitable for fully studying topological entanglement during replication.

Based on the referee's feedback, we have now added a note on strand passage into the Main text (Subsection "Simulations of a replicating chromosome", page 4) as well as into the Methods and SI. Furthermore, we have adjusted our discussion to better explain that whereas our steady-state simulations show the directions of equilibrium forces, the dynamic simulations show that topo-entropic segregation forces can be relevant even when the polymer is out of equilibrium.

- Is it possible that the main benefit of loop extruders in segregation is to guide kinetics, prevent traps and entanglements during genome replication, rather than slight modification to equilibrium entropic forces? Please discuss.

This is an interesting idea, and we cannot rule out that this plays a role *in vivo*. Because we can largely understand the effect of the presence of loop-extruders in our dynamic simulations based on a qualitative modification of the entropic forces due to a global change in chromosome topology, we believe that this is the dominant effect.

To assess whether active loop-extruder dynamics are crucial for segregation in our simulations, we ran an additional dynamic simulation with stationary loop-extruders tying together the chromosomal arms. Briefly, we let a 1D loop-extrusion simulation on a single chromosome reach a steady state, and used the last loop-extruder configuration to constrain an unreplicated chromosome. These links were kept fixed during a replication process. To ensure that also the newly replicated chromosome would be constrained by loop-extruders, whenever two monomers i, j bound by a loop-extruder were both replicated, their copies i', j' were also tied together with a loop-extruder.

Chromosomes from these simulations with fixed loop-extruders also show clearly enhanced segregation compared to the no loop-extruders case, although segregation is somewhat delayed (Fig. R2). This suggests loop-extruders can contribute to segregation by zipping up the chromosomal arms, and not only by an active mechanism that would prevent kinetic traps or entanglement on replicating chromosomes. We included the figure here, but decided not to include this data in our paper as the model with static loop-extruders is not physiologically relevant.

- Fig 2a,b. Is this a result of a single simulation? No error bars are shown. How replicable are these results? Please add error bars, or at least somehow indicate what is the statistical significance between the two curves.

Figure R2: **Dynamic simulations with stationary loop-extruder links.** **A** The segregated fraction over time in simulations with loop-extruders at fixed points. **B** The mean long-axis positions a quarter, half-way, three-quarters and at the end of the replication process.

Figure R3: **The mean long axis positions for steady-state simulations, with error bars showing the standard error on the mean.** The averages were taken over multiple simulations and over time. The error bars are narrower than the lines indicating the means: their width can be seen between the dashed lines.

Figure 2 A,B show the mean long axis positions of loci, averaged over time and over several simulations. We have adjusted the y-axis label to further emphasize this point, and a supplementary table of how many simulation runs were performed for each set of simulation parameters. As suggested by the referee, we also tried adding error bars that show the standard errors on the mean. However, these error bars are too small to be visible (Figure R3), which demonstrates that the difference between the mean curves is certainly significant.

- Fig 3C, why does the segregated fraction initially decrease without loop extruders?

Thanks for bringing this to our attention. This question allowed us to identify a minor bug in our code. Simply put, for the replicating simulations depicted in Figure 3, the replication fork positions vary between simulations. This means that at these early replication times, some chromosomes are unreplicated. These unreplicated configurations were unintentionally assigned a replicated fraction of 1, which also led to the mean value being high. We corrected the code by assigning unreplicated chromosomes segregated fractions of 0, and these corrected plots no longer show the initial decrease referred to by the referee (Fig. R4).

- Discussion section: Here authors state: “dynamic simulations reveal that purely entropic forces hardly contribute”. If this is the case, why spend the first half of the manuscript discussing entropic forces? Clarify whether the main role of loop extruders is to modify kinetics or thermodynamic forces for segregation.

We thank the referee for this feedback, and adjusted the text to better emphasize the significance of our work considering earlier theories in the field. Jun and Mulder (2006) proposed that purely entropic forces would be sufficient to cause bacterial chromosome segregation, and this result has been a widely cited in the field of bacterial chromosome segregation. However, this claim has also been questioned because bacteria are known to rely on active segregation mechanisms to achieve chromosome segregation. The first half of our manuscript hence focuses on disproving the claim that entropy alone could drive bacterial chromosome segregation on theoretical grounds: we show that at intermediate replication stages, entropic forces cannot drive bacterial chromosome segregation. Since bacterial chromosome segregation is concurrent with replication, this shows that even if there would be sufficient time for the chromosome to fully relax, entropy alone cannot explain bacterial chromosome segregation. This is important, as it resolves a long-standing debate.

The main role of loop-extruders in our model is to change the effective large-scale topology of the bacterial chromosome from circular to a zipped-up linear one. This effective linearization ties together the two arms of the chromosome, so that the entropically preferred state at intermediate replication stages now becomes segregated rather than fork-segregated. Our steady-state simulations prove this claim; at intermediate replication stages, the entropically preferred state is unsegregated in the absence of loop-extruders, and segregated in their presence. As the referee correctly points out (see also our response to the kinetics question), in the replicating simulations the chromosome is not at equilibrium. In this dynamic case, one can still say that the loop-extruders can redirect thermodynamic forces so that they would be driving

Figure R4: **Corrected mean segregated fraction for the replicating simulations.**

segregation. Thus, we argue that the main role of loop-extruders is to modify thermodynamic forces for segregation. We have now included a statement in the discussion to emphasize this point.

- Later in the discussion section states: “in our model, we employed non-topological loop extruders” The model is called topo-entropic and the title claims topology is considered. Related to a point raised above, clarify exactly which genome/loop-extruder topologic constraints are taken into account in the model and which are not.

We apologize for the confusion. In the field of chromosome organization, “non-topological” loop-extruders are loop-extruders capable of by-passing obstacles and each other upon collision, contrasting “topological” loop-extruders that would enclose DNA in a ring and would hence be incapable of by-passing each other and obstacles larger than the ring size. As noted by the reviewer, we use “non-topological” loop-extruders in our simulations, which are capable of by-passing each other, as observed both *in vitro* and *in vivo*. We have clarified this point in our manuscript, for instance by removing phrases such as “topological constraints imposed by loop-extruders”.

We hope that our revised manuscript better communicates that the title of our manuscript refers to our central claim that loop-extruders can contribute to bacterial chromosome segregation by changing the effective topology of the circular chromosome into a linear one. This is the key topological change that actually redirects entropic forces in our model.

- Direct quantitative comparison to existing experimental data (if they exist) would strengthen the claims presented.

As of yet, no published quantitative data is available to which we could directly compare statistics such as the mean long-axis positions of monomers throughout the replication cycle or the segregated fraction. We hope that our theoretical results will inspire various labs to measure such data. However, motivated by the referee’s suggestion, we have now included a comparison of the contact maps in our replicating simulations to existing Hi-C data for *C. crescentus* (Le et al., 2013). Hi-C experiments are used to measure how often different loci are spatially proximate, or “in contact”. We analyzed and normalized our simulation data analogously to experimental Hi-C data to enable a direct comparison (Figure R5).

The experimental contact maps show an off-diagonal line throughout the replication cycle, indicating interactions between the two chromosomal arms. For mutant strains without loop-extruding SMCs, this line be-

comes less dominant (Le et al., 2013). Comparing to our simulation data, we see that without loop-extruders (Columns 1 and 2 in the figure), the off-diagonal line disappears from the simulated contact maps over time. However, with loop-extruders (Columns 3 and 4), this line is maintained throughout the replication process. We also note that at early replication stages (second row), our simulation model with both loop-extruders and *ori*-pulling best matches the experimental data. In addition to showing the off-diagonal line mediated by loop-extruders, this map shows increased contacts between the *ori* and *ter* regions (indicated by yellow arrows in the figure). The good comparison with this model is in line with the well-established role of the *ori*-pulling ParABS system in *C. crescentus* chromosome segregation (Toro et al., 2008).

Our model makes many simplifications and therefore does not exactly match the Hi-C data. For instance, when loop-extruders collide with transcription machinery, they are known to slow down Brandão et al. (2019), which affects both the shape and spread of the condensin-mediated off-diagonal line on Hi-C maps. By comparison, in our model loop-extruders move at constant rates, which leads to less spread of the off-diagonal line. Additionally, our model does not exhibit box-like CID structures near the main diagonal of the contact map. CID boundaries are known to correspond to highly transcribed genes Le and Laub (2016), and since our model does not include transcription, we do not reproduce these patterns on the simulated contact maps.

We thank the referee for their suggestion, and feel that this direct comparison to experimental data indeed strengthens our manuscript. Because of length constraints, we have added this comparison as a new Supplementary Figure 13.

Reviewer 3 (Remarks to the Author):

Harju et al. report on their theoretical analysis and modeling of bacterial chromosome segregation to resolve the roles of three factors that have been proposed to assist in or drive segregation: chromosome configurational entropy, loop-extruding SMC complexes, and dedicated/directed forcing mechanisms, such as the ParABS system. In contrast to previous models, the authors consider all of these mechanisms together and their possible complementary roles, as well as the complication of chromosome replication while segregation proceeds. They find that although entropy of ring polymer chromosomes can in principle lead to segregation, this is not the ideal kinetically because segregation would occur more slowly than replication, and moreover, segregation is not the favored equilibrium state until after replication has been completed. Loop extrusion that ‘zips’ together chromosomal arms, given the model proposed by the authors, is sufficient to rescue an entropic mechanism and leads to more robust segregation outcomes. Similarly, ParABS-like mechanisms can segregate sister oris, but without chromosome compaction/‘linearization’ via loop extrusion, they are deficient. The authors argue that their model can explain the importance of origin-loaded loop extruders and the notable “secondary diagonal” observed in Hi-C experiments in bacteria.

Altogether, this is a very well considered effort to understand seemingly disparate mechanisms/factors of chromosome segregation. The work addresses valuable questions about the importance and interplay of these mechanisms, and uses simple, but fundamental and largely convincing physical arguments to do so. These simple theoretical arguments are then supported by extensive simulation work. From a technical standpoint, the authors excelled in developing simulations that could properly interface the various factors they considered. I also appreciated the model of the replicating chromosome, and it is the first of its kind of which I am aware. However, in places the authors were extremely terse where additional explanation would be helpful. Recognizing, that word count may be the issue, I generally think the manuscript reads well within these constraints, but there may be room for improvement. I also have a few questions about the choice of extrusion model in the context of replication given recent results suggesting that even non-topologically bound SMC complexes can be blocked by replication and transcription machinery. Altogether, this largely well written and well considered manuscript provides a good conceptual advance for the fields of chromosome biophysics and bacterial biology/biophysics.

We thank the referee for their evaluation and appreciation of our work. Also, we are grateful for the referee’s constructive feedback on further clarifying certain points in our manuscript, and made several improvements based on their suggestions below.

Figure R5: **Comparison of simulation data to Hi-C data from synchronized *C. crescentus* cells.** Rows correspond to 0, 10, 30, 45, 60 and 75 minutes after the start of replication. In each contact map, the upper-left triangle shows experimental data, and the lower-right triangle data from our simulations. Red dashed lines show expected replication fork positions. Columns correspond to simulations with neither loop-extruders nor *ori*-pulling, only *ori*-pulling, only loop-extruders, and both loop-extruders and *ori*-pulling. Yellow arrows indicate faint lines corresponding to contacts between the *ori* and *ter* regions, only reproduced in our model with both loop-extrusion and *ori*-pulling.

Figure R6: **Mean fork separations and positions with or without loop-extruders.** **A** The steady-state mean long axis separation between the replication forks, with or without loop-extruders. Line shows simple calculation for expected separation in the absence of loop-extruders, assuming that the separation scales with the length of the linear segment connecting the forks. Turquoise background shows the cell length at a given time-point. Ribbons show standard deviation. **B** Mean steady-state replication fork positions in simulations with loop-extruders. **C** Mean steady-state replication fork positions in simulations without loop-extruders.

Major comments: 1. Figure 2 and the main text describing it is difficult to follow. a) It might be helpful to readers to add markers for replication forks to plots like Fig 2A or at least directly stating that the fork is at the convergence of the orange, blue, and black lines.

We agree with the referee that identifying the replication forks in our figures is important. We have therefore made efforts to include small red spheres to indicate the positions of replication forks in all of our figures.

b) The authors may want to more clearly explain why different metrics are used to evaluate different scenarios in Fig 2E-F.

Thanks for this feedback. Indeed, Figures 2 E and F use different metrics for simulations without loop-extruders and with loop-extruders. This is because we expect the chromosomes to adopt qualitatively different configurations in the absence or presence of loop-extruders: without loop-extruders, we expect fork-segregated states, which should be characterized by a notable separation between the replication forks. In the presence of loop-extruders, on the other hand, the forks should be close to each other, since the chromosomal arms have been tied together. The distance between the forks is hence interesting for the case without loop-extruders, but shows a relatively flat line with loop-extruders (Figure R6A).

For the simulations with loop-extruders, we chose to study the position of the replication forks (Figure R6B), since our theory suggested that at half-way through replication, the forks should localize at roughly mid-cell, indicating a segregated *ori-ter-ori* configuration. If the referee is interested, we here include a similar plot for the replication fork positions without loop-extruders (Figure R6C), consistent with our fork separation data (Figure R6A).

In response to the referee's comments, we have made the following edits:

1. We include the mean fork separation in the presence of loop-extruders in Figure 2E.
 2. We edited the main text describing these subfigures and measures.
 3. We included the mean positions of both replication forks in Figure 2F for completeness.
 4. We changed the color schemes of the figures, based on other feedback from the referee.
- c) The colors, labeling of colors, and shading in Fig 2E-F are also confusing. Confinement length is gray,

Figure R7: **Segregated fractions for the replication factory model.** **A** Segregated fractions for steady state simulations with or without constraints tying the replication forks together. The yellow curve corresponds to a model with constraints on the forks, but no excluded volume interactions. **B** Segregated fractions from dynamic simulations of a replicating chromosome, with or without a replication factory model.

but why is it a region not a line? Turquoise is estimated maximum separation — is that the dark gray or the light blue? and why are these regions instead of lines?

We apologize for the confusion. We have now adjusted subfigures 2E and F so that the turquoise band indicates the cell length, consistent with Figure 2A. We hope that this clarifies the figure.

d) What does Fig 2G look like for the replication factory model?

We have included new supplementary figures 7E and F that show the segregated fraction for the replication factory model for both steady state and dynamic simulations (Figure R7). The segregated fraction of the replication factory model appears to be slightly worse, and the two values only converge as $R \rightarrow N$. This limit corresponds to the fully replicated case, when fork constraints no longer matter. For dynamic simulations, the segregated fraction curves with or without replication factory constraints look very similar, although the mean long axis position curves in Supplementary Figure 7D slightly differ.

2. It is argued that loop extrusion “linearizes” the chromosome arms. a) the meaning of this statement is not fully explained in the main text. It becomes clearer when the idea is revisited late in the manuscript or upon a close look at the supplemental scaling argument, but a few additional words here would help readers.

We thank the referee for this helpful feedback. We have adjusted the main text to better explain this statement.

b) what is the threshold for having “enough” loop extruders to “linearize” the chain? given ~ 40 SMCs (or maybe just a few!), it’s not obvious that the ‘zipped’ chromosome isn’t more like a few connected rings rather than a linearized chain.

The referee raises an interesting question. In the previous version of our manuscript, we halved and doubled the number of loop-extruders in our simulations, and found that 20 loop-extruders per chromosome still appeared to be sufficient for segregation. On the other hand, our replication factory simulations, which essentially give rise to three rings connected at the forks, did not give rise to segregation. The latter simulations suggest that tying together the replication forks does not prevent one newly replicated chromosomal strand from spreading out along the long axis of the chromosome. However, adding even one or two loop-extruders might already decrease such spreading, which suggests that even a small number of loop-extruders could contribute to bacterial chromosome segregation, as the referee points out.

To test this hypothesis, we ran new dynamic simulations with only 5 loop-extruders per chromosome. At these low values, segregation is significantly inhibited compared to having 20 or more loop-extruders (Figure R8).

Figure R8: **Further decrease of number of loop-extruders.** **A** Mean long-axis positions of monomers sampled at various time-points in dynamic simulations. There is notable enhancement of segregation when more than 20 loop-extruders per chromosome are used. **B** The mean segregated fractions for different loop-extruder numbers as a function of time in dynamic simulations.

However, there still appears to be some enhancement of segregation compared to the no loop-extruders case. We have added this new data to our revised manuscript, and hope that this answers the referee’s question.

c) It is hypothesized that the “improvement of segregation by loop-extruders could be due to repulsion between chromosomal loops caused by off-target loop-extruder loading”. I thought it was argued that “linearization” by extruders is responsible for segregation improvements. Furthermore, the hypothesis can be directly tested by the authors by simulating without off-target loading.

We thank the referee for this feedback. This sentence referred to the level of segregation at final replication stages. As proposed by Jun and Mulder (2006), in the limit of two ring polymers, entropic segregation should occur, and we hence expected that steady-state simulations with $R \approx N$ should show similar levels of segregation with or without loop-extruders. Surprisingly, however, we found that segregation was still better in the presence of loop-extruders. With this comment we wanted to offer an explanation in terms of an additional mechanism caused by off-target loop-extruder loading. For the scenario we consider in this manuscript, however, we argue that the linearization effect is the dominant mechanism.

As suggested by the referee, we tested our hypothesis by performing steady-state simulations at final replication stages with different levels of off-target loading. Indeed, increasing the amount of off-target loading by reducing the relative affinity to load at the *ori*, while keeping the number of loop-extruders constant, improves segregation at these final replication stages (Figure R9A), although the segregated fraction still remains slightly above the no loop-extruder case even with 99% targeted loading.

Additionally, we ran simulations to test that our main conclusions hold without off-target loading. First, we ran dynamic simulations, and found that the segregated fraction with targeted loop-extrusion is still significantly improved compared to without loop-extrusion (Figure R9C). We note that in this simplified scenario the segregated fraction starts to increase roughly half-way through replication, which is in-line with

our theoretical model.

We also found that with more specific loop-extruder loading, collisions between loop-extruders and replication forks become more frequent, and consequentially, the fraction of loop-extruders that unbinds at the replication forks, P_U , becomes increasingly important (Figure R9D).

Finally, we ran steady-state simulations with $P_U = 1$ (to prevent excessive loop-extruder links between sister strands) and 99% specific loading. As predicted by our topo-entropic segregation model, specifically loaded loop-extruders significantly improve segregation compared to the case without loop-extruders from half-way through replication (Figure R9B).

There is evidence based on prior work (Brandão et al., 2021; Wilhelm et al., 2015) that in living bacteria there will be both on and off-target loading, which led us to consider this scenario in the main text. Nonetheless, we agree with the reviewer that it is insightful to consider the only on-target loading scenario and have added these additional simulation results as Supplementary Note 8 and Supplementary Figure 10 into our manuscript. Finally, we note that our group is currently studying the effects of non-targeted loop-extruder loading on bacterial chromosome organization and segregation, and we feel that a longer discussion of this topic falls outside the scope of this manuscript.

3. The authors consider models where extrusion is either: 1) ‘non-topological’ so extrusion bypass replication forks and effectively jump to a new strand or engulf the fork or 2) ‘topological’ so the SMC encloses a replicated strand after encountering a replication fork. They also consider these models in the scenario where fork encounters facilitate SMC unloading.

But several recent papers argue that loop extruders might be pushed or otherwise obstructed by other complexes such as those involved in replication or transcription (ref 42 in this manuscript + Dequeker et al. Nature 2022, Jeppsson et al. Sci Adv 2022, Banigan et al. PNAS 2023). How might this alter the results?

The referee raises an important and subtle point. In our simplified model, we do not consider transcription, which has previously been modeled as slowing down SMCs as they collide with RNA polymerases (Brandão et al., 2019). Although such slowing down might cause slight changes in the SMC dynamics, this previous work suggests it does not have a significant effect on the “zipping up” of the chromosomal arms, which is the key ingredient for our model.

As the referee notes, we do coarsely model different possible interactions between replication forks and loop-extruders might affect our results. Lacking direct evidence for what happens during SMC replication fork collisions, we considered a model where upon such collisions, the SMC is either off-loaded with probability P_U or jumps to either strand on the other side of the fork, as expected if the fork acts as a permeable boundary.

We did not run simulations where replication forks would obstruct or push loop-extruders forward, but having considered the referee’s question, we can explain what we would expect in these scenarios.

First, we consider a pushing scenario. Since both loop-extruders and replication forks start off at the origin of replication, most collisions between these factors arise when replication forks “catch up” with a loop-extruder ahead of them. Including a pushing mechanism would result in SMCs being pushed ahead of the replication forks, giving rise to an effective replication factory mechanism. This is not expected to inhibit segregation as long as some loop-extruders would still be free to tie together the rest of the chromosomal arms, preventing the “arm spreading” effect we saw in our simulations with *only* a replication factory mechanism.

Second, if an obstruction mechanism was included, this would mainly impact SMCs that collide head-on with replication forks. In bacteria with targeted-loading of loop-extruders, this would mainly impact the size of the loops caused by non-targeted loading. As explained in our answer to the previous question 2C, this might inhibit segregation to some extent, since off-target loading of loop-extruders somewhat enhances segregation.

In response to the referee’s question, we have made the following additions to our manuscript:

1. We have added a sentence that states that we ignore slowing-down effects due to transcription, since

Figure R9: **Effect of loading specificity.** **A** Mean segregated fraction in steady-state simulations with $R \approx N$ for varying loading specificity. Error bars indicate standard deviation. **B** Mean segregated fractions for steady-state simulations with 99% specific loading and $P_U = 1$, and with no loop-extruders. **C** Segregated fraction from dynamic simulations with increased loading specificity. Affinity 4040 corresponds to results shown in the main text. Affinities 40400 and 404000 correspond to roughly 90% and 99% specific loading. **D** Segregated fraction with 99% specific loading, with either unbinding at the replication forks ($P_U = 1$) or no enhanced unbinding at the forks ($P_U = 0$).

based on previous work by Brandão et al. (2019), this should not affect the zipping up of the chromosomal arms.

2. We now mention the works cited by the referee on interactions between cohesin and replication machinery in eukaryotes.

Minor comments: 1. A summarizing sentence or two at the end of the 2nd results subsection “entropy does not segregate...” may improve clarity. As I understand it, the equilibrium ordering of oris and ters arises from maximal elongation of the largest rings, which differs between the first and second halves of replication, but somehow this logic doesn’t come through as clearly as it could.

Thanks for this feedback. We have adjusted the last paragraph of this section, and included a few sentences that summarize our key result and its implication: partially replicated chromosomes can maximize their entropy when both the largest ring and the linear segment are extended. This leads to unsegregated states where the replication forks are pushed apart being entropically preferred. This result implies that, unlike previously argued, purely entropic forces cannot drive bacterial segregation concurrent with replication.

2. please check supp fig references. e.g., rep factory is Fig S7, not S11 as stated in text fig s7.

Thanks for pointing out these errors, which we have fixed it in our revised manuscript.

3. The color key in Fig. 3C is a little confusing because of the reuse of gray and it being left to the reader to decipher the meaning of black/black outlines.

Thanks, we have adjusted the color scheme in our revised manuscript.

4. It would be valuable to include a supplemental table of major simulation parameters.

Following the referee’s suggestion, we have now included a table of our used parameters in the SI.

5. Does this work offer any insights about how entropy and other factors could segregate chromosomes in spherical bacteria?

This is certainly an exciting question. In principle, the zipping up of chromosomal arms by SMCs in spherical bacteria could also be beneficial for segregation, as it could still prevent potential fork segregation. However, an outstanding question for cocci is how the division plane is selected, and how the chromosome “knows” to segregate across this specific plane. These are questions that fall outside our current scope, but suggest avenues of future research.

6. In the supplement it is stated that “It cannot be assumed that the free energy cost per blob is the same for single chromosomal strands and doubled-up strands; since the doubled-up strand has more degrees of freedom, its free energy cost per blob is expected to be higher.” This is a confusing since blobs are usually defined by kT ! I guess it doesn’t matter since there are more higher energy blobs, so the argument holds, but the authors might want to clarify.

We thank the referee for this thoughtful comment. It is true that in polymer physics, confinement blobs are explicitly defined in terms of $k_B T$. However, the point that we wanted to make is that our blob argument relies on modeling the zipped-up region of the chromosome as a single effective chain, but in reality these are two polymer strands connected by loop-extruders at points. The doubled-up chain hence has many degrees of freedom in addition to the effective chain’s configuration. If we suppose that constraining the effective chain has the usual entropic cost of $1 k_B T$ per blob, if confinement additionally constraints any of these additional degrees of freedom, the total entropic cost of confinement would be slightly higher.

We have adjusted this section of the SI to further emphasize that modeling the zipped-up region as a single chain is an approximation, and that considering the additional degrees of freedom we neglect could increase the free energy cost of confinement. As the referee notes, this does not affect the outcome of our argument, since this additional entropic cost would only further disfavor fork-segregated states in the presence of loop-extruders.

References

- Suckjoon Jun and Bela Mulder. Entropy-driven spatial organization of highly confined polymers: Lessons for the bacterial chromosome. *Proc. Natl. Acad. Sci. U.S.A.*, 103(33):12388–12393, August 2006. doi: 10.1073/pnas.0605305103.
- Hugo B Brandão, Payel Paul, Aafke A van den Berg, David Z Rudner, Xindan Wang, and Leonid A Mirny. Rna polymerases as moving barriers to condensin loop extrusion. *Proceedings of the National Academy of Sciences*, 116(41):20489–20499, 2019.
- Hugo B. Brandão, Zhongqing Ren, Xheni Karaboja, Leonid A. Mirny, and Xindan Wang. DNA-loop-extruding SMC complexes can traverse one another in vivo. *Nat. Struct. Mol. Biol.*, 28(8):642–651, Aug 2021. ISSN 1545-9985. doi: 10.1038/s41594-021-00626-1.
- Tung B.K. Le, Maxim V. Imakaev, Leonid A. Mirny, and Michael T. Laub. High-resolution mapping of the spatial organization of a bacterial chromosome. *Science*, 342(6159):731–734, 2013. ISSN 10959203. doi: 10.1126/science.1242059.
- Esteban Toro, Sun-Hae Hong, Harley H. McAdams, and Lucy Shapiro. Caulobacter requires a dedicated mechanism to initiate chromosome segregation. *Proc. Natl. Acad. Sci. U.S.A.*, 105(40):15435–15440, October 2008. doi: 10.1073/pnas.0807448105.
- Tung B K Le and Michael T Laub. Transcription rate and transcript length drive formation of chromosomal interaction domain boundaries. *EMBO*, 35(14):1582–1595, 2016. ISSN 1460-2075.
- Larissa Wilhelm, Frank Bürmann, Anita Minnen, Ho-Chul Shin, Christopher P. Toseland, Byung-Ha Oh, and Stephan Gruber. SMC condensin entraps chromosomal DNA by an ATP hydrolysis dependent loading mechanism in *Bacillus subtilis*. *eLife*, May 2015. doi: 10.7554/eLife.06659.

REVIEWERS' COMMENTS

Reviewer #1 (Remarks to the Author):

The authors have addressed all my comments, and I recommend accepting this revised manuscript.

Reviewer #1 (Remarks on code availability):

I reviewed the code but didn't try to install it and reproduce the analysis myself.

The code is readable, and as I have some experience with the same chromosome simulation framework (polychrom), I assess that the code fits the description of the simulation in the manuscript. and a clear readme file and installation instructions are provided.

Reviewer #2 (Remarks to the Author):

The authors have addressed the comments and significantly clarified and improved the manuscript. I have no further comments. Nice work!

Reviewer #3 (Remarks to the Author):

The authors have thoroughly addressed my comments, and as far as I can tell, the other reviewers' comments. This manuscript nicely addresses long standing questions about the interplay between various chromosome segregation mechanisms in bacteria with both physical theory and computer modeling. The research significantly improves our overall understanding of the function of loop extrusion in bacteria and how it complements other mechanisms. I have two remaining minor, but non-critical suggestions. Overall I support publication.

1) I suggest the authors state in the figure legends (e.g., fig 2e-f) that the shaded regions around the data points correspond to standard deviation, or whatever it is.

2) On page 4 where the authors discuss fork-SMC interactions, the authors may want to more directly state that the $P_U > 1$ case is a kind of proxy for the replication machinery blocking extrusion (or else, that $P_U > 0$ is similar to at least partial extruder blockage).

Reviewer #1 (Remarks to the Author):

The authors have addressed all my comments, and I recommend accepting this revised manuscript.

We thank the referee for their time and feedback.

Reviewer #1 (Remarks on code availability):

I reviewed the code but didn't try to install it and reproduce the analysis myself. The code is readable, and as I have some experience with the same chromosome simulation framework (polychrom), I assess that the code fits the description of the simulation in the manuscript, and a clear readme file and installation instructions are provided.

Thanks for taking the time to check our code.

Reviewer #2 (Remarks to the Author):

The authors have addressed the comments and significantly clarified and improved the manuscript. I have no further comments. Nice work!

Thank you for the suggestions that helped us improve our manuscript.

Reviewer #3 (Remarks to the Author):

The authors have thoroughly addressed my comments, and as far as I can tell, the other reviewers' comments. This manuscript nicely addresses long standing questions about the interplay between various chromosome segregation mechanisms in bacteria with both physical theory and computer modeling. The research significantly improves our overall understanding of the function of loop extrusion in bacteria and how it complements other mechanisms. I have two remaining minor, but non-critical suggestions. Overall I support publication.

Thank you for helping us improve the manuscript, as well as supporting publication of our work.

1) I suggest the authors state in the figure legends (e.g., fig 2e-f) that the shaded regions around the data points correspond to standard deviation, or whatever it is.

We have now added these statements into the figure legends.

2) On page 4 where the authors discuss fork-SMC interactions, the authors may want to more directly state that the $P_U > 1$ case is a kind of proxy for the replication machinery blocking extrusion (or else, that $P_U > 0$ is similar to at least partial extruder blockage).

We have added a brief explanation that increasing P_U effectively stops loop extrusion at the replication forks.